# SUPERVISED CONTRASTIVE REGRESSION WITH SAMPLE RANKING

## ABSTRACT

Deep regression models typically learn in an end-to-end fashion and do not explicitly try to learn a regression-aware representation. Their representations tend to be fragmented and fail to capture the continuous nature of regression tasks. In this paper, we propose *Supervised Contrastive Regression (SupCR)*, a framework that learns a regression-aware representation by contrasting samples against each other based on their target distance. SupCR is orthogonal to existing regression models, and can be used in combination with such models to improve performance. Extensive experiments using five real-world regression datasets that span computer vision, human-computer interaction, and healthcare show that using SupCR achieves the state-of-the-art performance and consistently improves prior regression baselines on all datasets, tasks, and input modalities. SupCR also improves robustness to data corruptions, resilience to reduced training data, performance on transfer learning, and generalization to unseen targets.

## 1 INTRODUCTION

Regression problems are ubiquitous and fundamental in the real world. They include estimating age from human appearance (Rothe et al., 2015), predicting health scores from physiological signals (Engemann et al., 2022), and detecting gaze directions from webcam images (Zhang et al., 2017b). Since regression targets are continuous, the most widely used approach to train a regression model is to have the model predict the target value, and use the distance (e.g., L1 or L2 distance) between the prediction and the ground-truth target as the loss function (Zhang et al., 2017a;b; Schrumpf et al., 2021; Engemann et al., 2022). There are also works that control the relationship between predictions and targets by converting the regression task into a classification task and training the model with the cross-entropy loss (Rothe et al., 2015; Niu et al., 2016; Shi et al., 2021).

However, all previous methods focus on imposing constraints on the final predictions in an end-to-end fashion, but do not explicitly consider the *representations* learned by the model. Their representations tend to be fragmented and fail to capture the continuous relationships underlying regression tasks. For example, Figure 1(a) shows the representation learned by the L1 loss in the task of predicting weather temperature from webcam outdoor images (Chu et al., 2018), where the images are captured by 44 outdoor webcams at different locations. The representation learned by the L1 model does not exhibit the continuous ground-truth temperatures; rather it is grouped by different webcams in a fragmented manner. Such unordered and fragmented representation is sub-optimal for the regression task and can even hamper performance since it contains distracting information (e.g., the capturing webcam).

While there is a rich literature on representation learning, past methods are focused on classification problems. In particular, contrastive learning (Chen et al., 2020a; He et al., 2020; Chen et al., 2020b) and supervised contrastive learning (SupCon) (Khosla et al., 2020) have been proven highly effective in representation learning. However, as shown in Figure 1(b), which plots the representation learned by SupCon for the visual temperature prediction task mentioned above, such method produces a sub-optimal representation for regression problems because it ignores the continuous order between the samples in a regression task. Besides, there are several recent works (Wang et al., 2022; Dufumier et al., 2021a;b; Schneider et al., 2022) adopting contrastive learning in the context of continuous labels, but they are not doing regression learning tasks.

In this paper, we introduce *Supervised Contrastive Regression (SupCR)*, a new framework for deep regression learning, where we first learn a representation that ensures the distances in the embedding

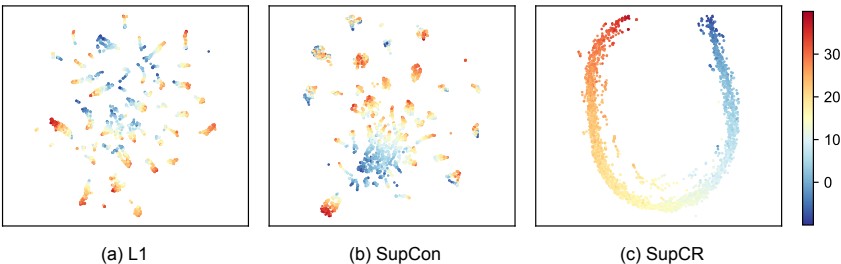

| (a) L1 | (b) SupCon | (c) SupCR |

Figure 1: UMAP (McInnes et al., 2018) visualization of the representation learned by L1, Sup-Con (Khosla et al., 2020), and the proposed SupCR for the task of predicting the temperature from webcam outdoor images (Chu et al., 2018). The representation of each image is visualized as a dot and the color indicates the ground-truth temperature. SupCR can learn a representation that captures the intrinsic ordered relationships between the samples while L1 and SupCon fail to do so.

space are ordered according to the target values. We then use this representation to predict the targets. To learn such a regression-aware representation, we contrast samples against each other based on their label/target distances. Our method explicitly leverages the ordered relationships between the samples to optimize the representation for the downstream regression task. As shown in Figure 1(c), SupCR leads to a representation that captures the intrinsic ordered relationships between the samples. Moreover, our framework is orthogonal to existing regression methods as one can use any kind of regression method to map our learned representation to the prediction values.

Extensive experiments using five real-world regression datasets that span computer vision, human-computer interaction, and healthcare verify the superior performance of our framework. The results show that SupCR achieves the state-of-the-art performance and consistently improves the performance of previous regression methods by **8.7%** on average across all datasets, tasks, and input modalities. Furthermore, our experimental results also show that SupCR delivers multiple desirable properties: 1) improved robustness to data corruptions; 2) enhanced resilience to reduced training data; 3) higher performance on transfer learning; and 4) better generalization to unseen targets.

## 2   RELATED WORK

**Regression Learning.**   Deep learning has achieved a great success in addressing regression tasks (Rothe et al., 2015; Zhang et al., 2017a;b; Schrumpf et al., 2021; Engemann et al., 2022). The most straightforward and widely used way to train a regression model is to have the model predict the target value, and use the distance between the prediction and the ground-truth target as the loss function. Common distance metrics used as the loss function in regression tasks are the L1 loss, the MSE loss and the Huber loss (Huber, 1992).

Past work has also proposed variants to the basic methods described above. One branch of prior works (Rothe et al., 2015; Gao et al., 2017; 2018; Pan et al., 2018) divide the regression range into small bins to convert the problem into a classification problem, and use classification loss, e.g., the cross-entropy loss, to train the model. Another branch of prior work (Niu et al., 2016; Fu et al., 2018; Cao et al., 2020; Shi et al., 2021) casts regression as an ordinal classification problem. They design multiple ordered thresholds and use multiple binary classifiers, one for each threshold, to learn whether the target is larger than each threshold.

Past work however focuses on forcing the final predictions of the model to be close to the target and does so in an end-to-end manner. In contrast, this paper focuses on ordering the samples in the embedding space in accordance with the label order, and it does so by first learning a representation that contrasts samples against each other, and then learning a predictor to map the representation to the prediction values. Therefore, our method is orthogonal to prior methods for regression learning as those prior methods can be applied to training our predictor and learning the final predictions with the help of our ordered representation.

**Contrastive Learning.**   Contrastive learning, which aligns positive pairs and repulses negative pairs in the representation space, has demonstrated improved performance for self-supervised learn-

ing (Chen et al., 2020a; He et al., 2020; Chen et al., 2020b; Chuang et al., 2020). Its supervised version, named supervised contrastive learning (SupCon) (Khosla et al., 2020), which defines samples from the same class as positive pairs and samples from different classes as negative pairs, was also shown to outperform the cross-entropy loss on image classification tasks (Khosla et al., 2020). It is also beneficial for learning in challenging settings such as noisy labels (Li et al., 2022a), long-tailed classification (Kang et al., 2020; Li et al., 2022b) and out-of-domain detection (Zeng et al., 2021).

A couple of recent papers have used the concept of "contrastive" or contrastive learning in the context of continuous labels. In particular, Yu et al. (2021) learns a model for action quality assessment by regressing the relative scores between two input videos. This work differs from ours in that it does not use the standard contrastive learning framework. Wang et al. (2022) proposes to improve domain adaptation for gaze estimation by adding a contrastive loss term to the L1 loss. They show that their approach is beneficial for adapting a gaze estimation model from one dataset (i.e., one domain) to another, but the approach produces no benefits and even reduces performance for the source dataset. In contrast, our approach improves performance for the source dataset as opposed to being specific to domain adaptation. Besides, Dufumier et al. (2021a;b) use a contrastive loss reweighted by continuous meta-data for classification. Schneider et al. (2022) learns low-dimensional and interpretable embeddings to encode behavioral and neural data using a generalized contrastive loss which samples positives and negatives according to continuous behavior or time labels. However, the continuous meta-data in (Dufumier et al., 2021a;b) and the continuous labels in (Schneider et al., 2022) are not the targets in their tasks and they are not working on regression problems.

## 3 METHOD

For a regression task, we aim to train a neural network composed of a feature encoder $f(\cdot) : X \to \mathbb{R}^{d_e}$ and a predictor $p(\cdot) : \mathbb{R}^{d_e} \to \mathbb{R}^{d_t}$ to predict the target $\boldsymbol{y} \in \mathbb{R}^{d_t}$ based on the input data $\boldsymbol{x} \in X$.

Given an input batch of data, similar to contrastive learning methods (Chen et al., 2020a; Khosla et al., 2020), we first apply data augmentation twice to obtain two views of the batch. The two views are fed into the encoder $f(\cdot)$ to obtain a $d_e$-dimensional feature embedding for each augmented input data. Our supervised contrastive regression loss, $\mathcal{L}_{\text{SupCR}}$, is computed on the feature embeddings. To use the learned representation for regression, we freeze the encoder $f(\cdot)$ and train the predictor on top of it using a regression loss (e.g., L1 loss).

### 3.1 SUPERVISED CONTRASTIVE REGRESSION LOSS

We would like our loss function to ensure distances in the embedding space are ordered according to distances in the label space. But, how can we design a contrastive loss that delivers this property? Below, we first define our loss function, then explain it in the context of positive and negative contrastive samples, and finally provide a theoretical analysis to prove that as our loss approaches its minimum, the learned representation will be ordered according to label distances.

For a positive integer $I$, let $[I]$ denote the set $\{1, 2, \cdots, I\}$. Given a randomly sampled batch of $N$ input and label pairs $\{(\boldsymbol{x}_n, \boldsymbol{y}_n)\}_{n \in [N]}$, we apply data augmentation to the batch and obtain a two-view batch $\{(\tilde{\boldsymbol{x}}_\ell, \tilde{\boldsymbol{y}}_\ell)\}_{\ell \in [2N]}$, where $\tilde{\boldsymbol{x}}_{2n} = t(\boldsymbol{x}_n)$, $\tilde{\boldsymbol{x}}_{2n-1} = t'(\boldsymbol{x}_n)$ ($t$ and $t'$ are independently sampled data augmentations operations) and $\tilde{\boldsymbol{y}}_{2n} = \tilde{\boldsymbol{y}}_{2n-1} = \boldsymbol{y}_n, \forall n \in [N]$. The augmented batch is fed into the encoder $f(\cdot)$ to obtain $\boldsymbol{v}_l = f(\tilde{\boldsymbol{x}}_l) \in \mathbb{R}^{d_e}, \forall l \in [2N]$. Our supervised contrastive regression loss is then computed over the feature embeddings $\{\boldsymbol{v}_l\}_{l \in [2N]}$ as:

$$\mathcal{L}_{\text{SupCR}} = -\frac{1}{2N} \sum_{i=1}^{2N} \frac{1}{2N-1} \sum_{j=1, \, j \neq i}^{2N} \log \frac{\exp(\text{sim}(\boldsymbol{v}_i, \boldsymbol{v}_j)/\tau)}{\sum_{k=1}^{2N} \mathbb{1}_{[k \neq i, \, d(\tilde{\boldsymbol{y}}_i, \tilde{\boldsymbol{y}}_k) \geq d(\tilde{\boldsymbol{y}}_i, \tilde{\boldsymbol{y}}_j)]} \exp(\text{sim}(\boldsymbol{v}_i, \boldsymbol{v}_k)/\tau)}, \quad (1)$$

where $\text{sim}(\cdot, \cdot)$ is the function measuring the similarity between two feature embeddings (e.g., negative L2 norm), $d(\cdot, \cdot)$ is the function measuring the distance between two labels (e.g., L1 distance), $\mathbb{1}_{[\cdot]} \in \{0, 1\}$ is an indicator function evaluating to 1 iff the conditions in the square brackets are satisfied, and $\tau$ is the temperature parameter.

At a high level, our loss can be explained in the context of positive/negative pairs, commonly used in contrastive learning. Contrastive learning and SupCon focus on classification. Hence, positive

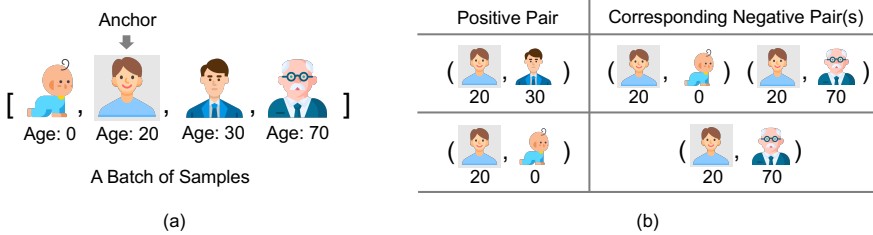

Figure 2: Illustration of $\mathcal{L}_{\text{SupCR}}$. (a) An example batch of input data and their age labels. (b) Two example positive pairs and the corresponding negative pair(s) when the anchor is the 20-year old man (shown in gray shading). When the 30-year old man creates a positive pair with the anchor, their label distance is 10, so the corresponding negative samples are the 0-year old baby and the 70-year old man, whose label distances to the anchor are 20 and 50 respectively. When the 0-year old baby creates a positive pair with the anchor, their label distance is 20. Only one sample in the batch has a larger label distance to the anchor, namely the 70-year old man, and it acts as a negative sample.

samples are those that come from the same class or the same input image, and all other samples are negative samples. In regression problems, there are no classes but continuous labels. Any two samples can be thought of as a positive pair or a negative pair, depending on context. To exploit the inherent continuity underlying the labels, we define positive and negative samples in a *relative* way, based on their label distance to the anchor. In particular, we rank samples with respect to their label distance to the anchor. We then contrast the anchor to the sample in the batch that is *closest* in the label space, which will be treated as the positive sample. All other samples are farther from the anchor and should be negative samples. We also contrast the anchor to the *second closest* sample in the batch. In this case, only samples in the batch that are farther from the positive sample (i.e., samples whose ranks is three or higher) can act as negative samples. We continue this process for higher rank samples (the third closest, fourth closest, etc.), and all anchors in a batch.

Figure 2(a) shows an example batch, and Figure 2(b) shows two positive pairs and their corresponding negative pair(s). Specifically, for any anchor sample $i$, *any* other sample $j$ in the same batch can be used to create a positive pair with corresponding negative samples set to all samples in the batch whose labels differ from $i$'s label by more than the label of $j$. In other words, $\mathcal{L}_{\text{SupCR}}$ aims to force the feature similarity between $i$ and $j$ greater than the feature similarity between $i$ and any other sample $k$ in the batch if the label distance between $i$ and $k$ is greater than $i$ and $j$. Therefore, optimizing $\mathcal{L}_{\text{SupCR}}$ will make the feature embedding ordered according to the order in the label space. The design of $\mathcal{L}_{\text{SupCR}}$ sufficiently leverages the ordered relationships between samples reflected in the continuous label space of regression problems, which is also the intrinsic difference between regression problems and classification problems.

## 3.2 Theoretical Analysis

In this section, we analytically show that optimizing $\mathcal{L}_{\text{SupCR}}$ will make the feature embedding ordered according to the order in the label space. To show this, we first derive a lower bound of $\mathcal{L}_{\text{SupCR}}$, and show $\mathcal{L}_{\text{SupCR}}$ can be arbitrarily close to it. Next, we formalize the concept of $\delta$-ordering which refers to the feature embeddings being ordered according to the order in the label space, and show that as $\mathcal{L}_{\text{SupCR}}$ gets sufficiently close to the lower bound, the feature embeddings will be $\delta$-ordered. All proofs are in Appendix A.

**Notations.** Let $s_{i,j} := \text{sim}(\boldsymbol{v}_i, \boldsymbol{v}_j)/\tau, \forall i, j \in [2N]$, $d_{i,j} := d(\tilde{\boldsymbol{y}}_i, \tilde{\boldsymbol{y}}_j), \forall i, j \in [2N]$, $D_{i,1} < D_{i,2} < \cdots < D_{i,M_i}$ be the sorted label distances starting from the $i$-th sample, i.e., $\text{sort}(\{d_{i,j} | j \in [2N] \backslash \{i\}\}), \forall i \in [2N]$, and $n_{i,m} := |\{j \mid d_{i,j} = D_{i,m}, \ j \in [2N] \backslash \{i\}\}|$ be the number of samples whose distance from the $i$-th sample equals $D_{i,m}, \forall\, i \in [2N], m \in [M_i]$.

First, we show that $L^\star := \frac{1}{2N(2N-1)} \sum_{i=1}^{2N} \sum_{m=1}^{M_i} n_{i,m} \log n_{i,m}$ is a lower bound of $\mathcal{L}_{\text{SupCR}}$.

**Theorem 1** (Lower bound of $\mathcal{L}_{\text{SupCR}}$). *$L^\star$ is a lower bound of $\mathcal{L}_{\text{SupCR}}$, i.e., $\mathcal{L}_{\text{SupCR}} > L^\star$.*

Next, we show that $\mathcal{L}_{\text{SupCR}}$ can be arbitrarily close to its lower bound $L^\star$.

**Theorem 2** (Lower bound tightness). *For any $\epsilon > 0$, there exists a set of feature embeddings such that $\mathcal{L}_{SupCR} < L^\star + \epsilon$.*

Then, we define a property called $\delta$-*ordered* for the feature embeddings $\{v_l\}_{l \in [2N]}$.

**Definition 1** ($\delta$-ordered feature embedddings). *For any $0 < \delta < 1$, the feature embeddings $\{v_l\}_{l \in [2N]}$ are $\delta$-ordered if $\forall i \in [2N], j, k \in [2N] \backslash \{i\}$,*

$$\begin{cases} s_{i,j} > s_{i,k} + \dfrac{1}{\delta} \ \text{if} \ d_{i,j} < d_{i,k} \\ |s_{i,j} - s_{i,k}| < \delta \ \text{if} \ d_{i,j} = d_{i,k} \\ s_{i,j} < s_{i,k} - \dfrac{1}{\delta} \ \text{if} \ d_{i,j} > d_{i,k} \end{cases} .$$

It says that a $\delta$-ordered set of feature embeddings satisfies the following properties: First, for all $j$ and $k$ such that $d_{i,j} = d_{i,k}$, the gap between $s_{i,j}$ and $s_{i,k}$ should be smaller than $\delta$; Second, for all $j$ and $k$ such that $d_{i,j} < d_{i,k}$, $s_{i,j}$ should be larger than $s_{i,k}$ by at least $\frac{1}{\delta}$. Notice that $\frac{1}{\delta} > \delta$, which means the feature similarity gap between a sample pair with different label distance to the anchor is always larger than the feature similarity gap between a sample pair with equal label distance to the anchor.

Finally, we show that for any $0 < \delta < 1$, when $\mathcal{L}_{SupCR}$ is close enough to its lower bound $L^\star$, the feature embeddings will be $\delta$-ordered.

**Theorem 3** (Main theorem). *For any $0 < \delta < 1$, there exist $\epsilon > 0$, such that if $\mathcal{L}_{SupCR} < L^\star + \epsilon$, then the feature embeddings are $\delta$-ordered.*

## 4 EXPERIMENTS

In this section, we evaluate our proposed method empirically. We first benchmark our method and compare it with the state-of-the-art regression baselines. Then, we evaluate the desirable properties of our learned representations, including the robustness to data corruptions, the resilience to reduced training data, the performance on transfer learning, and the generalization ability to unseen targets. Finally, we analyze and evaluate the design variants for our method. Refer to Appendix E for additional experiments and analysis.

**Datasets.** We benchmark our method on five regression datasets for common real-world tasks in computer vision, human-computer interaction, and healthcare. Ethics statements of the datasets are included in Appendix B.

1) AgeDB (Moschoglou et al., 2017) is used for predicting age from face images. It contains 16,488 in-the-wild images of celebrities and the corresponding age labels. The age range is between 0 and 101. It is split into a 12,208-image training set, a 2140-image validation set and a 2140-image test set.

2) TUAB (Obeid & Picone, 2016; Engemann et al., 2022) is used for brain-age estimation from EEG resting-state signals. The dataset comes from EEG exams at the Temple University Hospital in Philadelphia. Following Engemann et al. (2022), we use only the non-pathological subjects, so that we may consider their chronological age as their brain-age label. The dataset includes 1,385 21-channel EEG signals sampled at 200Hz from individuals whose age ranges from 0 to 95. It is split into a 1,246-subject training set and a 139-subject test set.

3) MPIIFaceGaze (Zhang et al., 2017a;b) is used for gaze direction estimation from face images. It contains 213,659 face images collected from 15 participants during their natural everyday laptop use. We subsample and split it into a 33,000-image training set, a 6,000-image validation set and a 6000-image test set with no overlapping participants. The gaze direction is described as a 2-dimensional vector with the pitch angle in the first dimension and the yaw angle in the second dimension. The range of the pitch angle is -40° to 10° and the range of the yaw angle is -45° to 45°.

4) SkyFinder (Mihail et al., 2016; Chu et al., 2018) is used for temperature prediction from outdoor webcam images. It contains 35,417 images captured by 44 cameras around 11am on each day under a wide range of weather and illumination conditions. The temperature range is $-20\,^\circ\mathrm{C}$ to $49\,^\circ\mathrm{C}$. It is split into a 28,373-image training set, a 3,522-image validation set and a 3,522-image test set.

Table 1: Evaluation results on `AgeDB`.

| Method | MAE$^\downarrow$ | R$^{2\uparrow}$ | Method | MAE$^\downarrow$ | R$^{2\uparrow}$ |
|---|---|---|---|---|---|
| L1 | 6.63 | 0.828 | SUPCR(L1) | **6.14** (+0.49) | **0.850** (+0.022) |
| MSE | 6.57 | 0.828 | SUPCR(MSE) | 6.19 (+0.38) | 0.849 (+0.021) |
| HUBER | 6.54 | 0.828 | SUPCR(HUBER) | 6.15 (+0.39) | **0.850** (+0.022) |
| DEX (Rothe et al., 2015) | 7.29 | 0.787 | SUPCR(DEX) | 6.43 (+0.86) | 0.836 (+0.049) |
| DLDL-v2 (Gao et al., 2018) | 6.60 | 0.827 | SUPCR(DLDL-v2) | 6.32 (+0.28) | 0.844 (+0.017) |
| OR (Niu et al., 2016) | 6.40 | 0.830 | SUPCR(OR) | 6.34 (+0.06) | 0.843 (+0.013) |
| CORN (Shi et al., 2021) | 6.72 | 0.811 | SUPCR(CORN) | 6.44 (+0.28) | 0.838 (+0.027) |

Table 2: Evaluation results on `TUAB`.

| Method | MAE$^\downarrow$ | R$^{2\uparrow}$ | Method | MAE$^\downarrow$ | R$^{2\uparrow}$ |
|---|---|---|---|---|---|
| L1 | 7.96 | 0.599 | SUPCR(L1) | 6.97 (+0.99) | **0.697** (+0.098) |
| MSE | 8.06 | 0.585 | SUPCR(MSE) | 7.05 (+1.01) | 0.692 (+0.107) |
| HUBER | 7.59 | 0.637 | SUPCR(HUBER) | 6.99 (+0.60) | 0.696 (+0.059) |
| DEX (Rothe et al., 2015) | 8.01 | 0.537 | SUPCR(DEX) | 7.23 (+0.78) | 0.646 (+0.109) |
| DLDL-v2 (Gao et al., 2018) | 7.91 | 0.560 | SUPCR(DLDL-v2) | **6.91** (+1.00) | **0.697** (+0.137) |
| OR (Niu et al., 2016) | 7.36 | 0.646 | SUPCR(OR) | 7.01 (+0.35) | 0.688 (+0.042) |
| CORN (Shi et al., 2021) | 8.11 | 0.597 | SUPCR(CORN) | 7.22 (+0.89) | 0.663 (+0.066) |

Table 3: Evaluation results on `MPIIFaceGaze`.

| Method | Angular$^\downarrow$ | R$^{2\uparrow}$ | Method | Angular$^\downarrow$ | R$^{2\uparrow}$ |
|---|---|---|---|---|---|
| L1 | 5.97 | 0.744 | SUPCR(L1) | 5.27 (+0.70) | 0.815 (+0.071) |
| MSE | 6.02 | 0.747 | SUPCR(MSE) | 5.35 (+0.67) | 0.802 (+0.055) |
| HUBER | 6.34 | 0.709 | SUPCR(HUBER) | 5.15 (+1.19) | **0.830** (+0.121) |
| DEX (Rothe et al., 2015) | 5.72 | 0.776 | SUPCR(DEX) | 5.14 (+0.58) | 0.805 (+0.029) |
| DLDL-v2 (Gao et al., 2018) | 5.47 | 0.799 | SUPCR(DLDL-v2) | 5.16 (+0.31) | 0.802 (+0.003) |
| OR (Niu et al., 2016) | 5.86 | 0.770 | SUPCR(OR) | **5.13** (+0.73) | 0.825 (+0.055) |
| CORN (Shi et al., 2021) | 5.88 | 0.762 | SUPCR(CORN) | 5.18 (+0.70) | 0.820 (+0.058) |

Table 4: Evaluation results on `SkyFinder`.

| Method | MAE$^\downarrow$ | R$^{2\uparrow}$ | Method | MAE$^\downarrow$ | R$^{2\uparrow}$ |
|---|---|---|---|---|---|
| L1 | 2.95 | 0.860 | SUPCR(L1) | 2.86 (+0.09) | **0.869** (+0.009) |
| MSE | 3.08 | 0.851 | SUPCR(MSE) | 2.86 (+0.22) | **0.869** (+0.018) |
| HUBER | 2.92 | 0.860 | SUPCR(HUBER) | 2.86 (+0.06) | **0.869** (+0.009) |
| DEX (Rothe et al., 2015) | 3.58 | 0.778 | SUPCR(DEX) | 2.88 (+0.70) | 0.865 (+0.087) |
| DLDL-v2 (Gao et al., 2018) | 2.99 | 0.856 | SUPCR(DLDL-v2) | **2.85** (+0.14) | **0.869** (+0.013) |
| OR (Niu et al., 2016) | 2.92 | 0.861 | SUPCR(OR) | 2.86 (+0.06) | 0.867 (+0.006) |
| CORN (Shi et al., 2021) | 3.24 | 0.819 | SUPCR(CORN) | 2.89 (+0.35) | 0.862 (+0.043) |

5) `IMDB-WIKI` (Rothe et al., 2015) is a large dataset for predicting age from face images, which contains 523,051 celebrities images and the corresponding age labels. The age range is between 0 and 186 (some images are mislabeled). We use this dataset to test our method's resilience to reduced training data, performance on transfer learning, and ability to generalize to unseen targets. We subsample the dataset to create a variable size training set, and keep the validation set and test set unchanged with 11,022 images in each.

**Metrics.** We report two kinds of metrics: the prediction error and coefficient of determination ($R^2$). Prediction errors have practical meaning and are convenient for interpretation, while $R^2$ quantifies how much the model outperform a dummy regressor that always predicts the mean value of the training labels. For age, brain-age, and temperature, the mean absolute error (MAE) is reported as the prediction error. For gaze direction, the angular error is reported as the prediction error.

**Baselines.** We implemented seven typical regression methods as baselines. L1, MSE and HUBER have the model directly predict the target value and train the model with an error-based loss function, where L1 uses the mean absolute error, MSE uses the mean squared error and HUBER uses an MSE

term when the error is below a threshold and an L1 term otherwise. DEX (Rothe et al., 2015) and DLDL-v2 (Gao et al., 2018) divide the regression range of each label dimension into several bins and learn the probability distribution over the bins. DEX (Rothe et al., 2015) optimizes a cross-entropy loss between the predicted distribution and the one-hot ground-truth labels, while DLDL-v2 (Gao et al., 2018) jointly optimizes a KL loss between the predicted distribution and a normal distribution centered at the ground-truth value, as well as an L1 loss between the expectation of the predicted distribution and the ground-truth value. During inference, they output the expectation of the predicted distribution for each label dimension. OR (Niu et al., 2016) and CORN (Shi et al., 2021) design multiple ordered threshold for each label dimension, and learn a binary classifier for each threshold. OR (Niu et al., 2016) optimizes a binary cross-entropy loss for each binary classifier to learn whether the target value is larger than each threshold, while CORN (Shi et al., 2021) learns whether the target value is larger than each threshold conditioning on it is larger than the previous threshold. During inference, they aggregate all binary classification results to produce the final results.

**Experiment Settings.** We use ResNet-18 (He et al., 2016) as the backbone model for `AgeDB`, `IMDB-WIKI`, `MPIIFaceGaze` and `SkyFinder` datasets. For `TUAB`, we use a 24-layer 1D ResNet (He et al., 2016) as the backbone model to process the EEG signals. We use the linear regressor as the predictor. We use the SGD optimizer and cosine learning rate annealing (Loshchilov & Hutter, 2016) for training. The batch size is set to 256. We pick the best learning rates and weight decays by grid search for both our method and the baselines. We train all the baselines and the encoder of our method for 400 epochs, and the linear regressor of our method for 100 epochs. The same data augmentations are adopted for all baselines and our method: random crop and resize (with random flip), color distortions for `AgeDB`, `IMDB-WIKI`, `SkyFinder`; random crop and resize (without random flip), color distortions for `MPIIFaceGaze`; random crop for `TUAB`. Appendix C shows augmentation examples for each dataset. Negative L2 norm, i.e., $\text{sim}(\boldsymbol{v}_i, \boldsymbol{v}_j) = -\|\boldsymbol{v}_i - \boldsymbol{v}_j\|_2$ is used as the feature similarity measure in $\mathcal{L}_{\text{SupCR}}$. L1 distance is used as the label distance measure in $\mathcal{L}_{\text{SupCR}}$ for `AgeDB`, `IMDB-WIKI`, `SkyFinder` and `TUAB`, while angular distance is used as the label distance measure for `MPIIFaceGaze`. The temperature $\tau$ is set to 2.0. Refer to Appendix D for more details.

## 4.1 MAIN RESULTS

As explained earlier, our model learns a regression-suitable representation that can be used by any of the baselines. Thus, in our comparison, we first train the encoder with the proposed $\mathcal{L}_{\text{SupCR}}$, and then freeze the encoder and train a predictor on top of it using each of the baseline methods. We then compare the original baseline without our representation, to the same baseline with our representation. For example, L1 means training the encoder and predictor end-to-end with L1 loss and SUPCR(L1) means training the encoder with $\mathcal{L}_{\text{SupCR}}$ and then training the predictor with L1 loss.

Table 1, Table 2, Table 3 and Table 4 show the evaluation results on `AgeDB`, `TUAB`, `MPIIFaceGaze` and `SkyFinder`, respectively. Green numbers highlight the performance gains brought by using our representation and the best numbers are shown in bold. As all tables indicate, SUPCR consistently achieves the best performance on both metrics across all datasets. Moreover, incorporating SUPCR to learn the representation consistently reduces the prediction error of all baselines by 5.8%, 10.1%, 11.7% and 7.0% on average on the `AgeDB`, `TUAB`, `MPIIFaceGaze` and `SkyFinder`, respectively.

## 4.2 ROBUSTNESS TO DATA CORRUPTIONS

It is well known that deep neural networks are highly vulnerable to out of distribution data and different forms of corruptions (Hendrycks & Dietterich, 2019), such as noise, blur and color distortions. Using the corruption generation process in ImageNet-C benchmark (Hendrycks & Dietterich, 2019), we corrupt the `AgeDB` test set with 19 diverse corruption types under different severity levels. In Figure 3, we compare SUPCR(L1) with L1 by testing the model (trained on clean `AgeDB` training set) on corrupted test set and reporting the average mean absolute error over all types of corruptions for each severity level. The re-

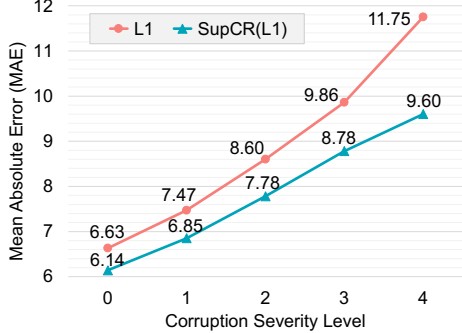

Figure 3: SUPCR is more robust to data corruptions. Figure shows MAE$^\downarrow$ as a function of corruption severity.

sults show that the representation learned by SUPCR is more robust to unforeseen data corruptions, and SUPCR suffers from lesser performance degradation as corruption severity increases.

## 4.3 RESILIENCE TO REDUCED TRAINING DATA

Much of the success of modern deep learning is due to the availability of massive training datasets. However, in many real-world settings, collecting a large training set is often infeasible due to the time and cost of labeling. Thus, it is desirable to improve model resilience to reduced training data. In Figure 4, we subsample IMDB-WIKI to produce varying size training sets, and compare SUPCR(L1) with L1. The results show that SUPCR is more resilient to reduced training data, and exhibits lesser performance degradation as the number of training samples decreases.

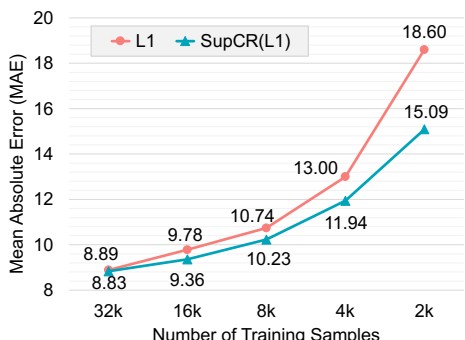

Figure 4: SUPCR is more resilient to reduced training data. Figure shows MAE$^\downarrow$ as a function of the number of training samples.

## 4.4 TRANSFER LEARNING

We evaluate the transfer learning performance by first pre-training the feature encoder on a large dataset and then using either linear probing (fixed encoder) or fine-tuning to learn a predictor on a small dataset. We investigate two scenarios: transferring from AgeDB which contains $\sim$12k samples to a subsampled IMDB-WIKI of 2k samples and transferring from another subsampled IMDB-WIKI of 32k samples to AgeDB. As shown in Table 5, SUPCR(L1) outperforms L1 in both the linear probing and fine-tuning settings in both two scenarios.

Table 5: Transfer learning results.

| Method | MAE$^\downarrow$ | R$^{2\uparrow}$ | Method | MAE$^\downarrow$ | R$^{2\uparrow}$ |
|---|---|---|---|---|---|
| AgeDB → IMDB-WIKI (subsampled, 2k): | | | | | |
| *Linear Probing:* | | | *Fine-tuning:* | | |
| L1 | 12.25 | 0.496 | L1 | 11.57 | 0.528 |
| SUPCR(L1) | 11.12 | 0.556 | SUPCR(L1) | 11.09 | 0.546 |
| | (+1.13) | (+0.060) | | (+0.48) | (+0.018) |
| IMDB-WIKI (subsampled, 32k) → AgeDB: | | | | | |
| *Linear Probing:* | | | *Fine-tuning:* | | |
| L1 | 7.36 | 0.801 | L1 | 6.36 | 0.848 |
| SUPCR(L1) | 7.06 | 0.812 | SUPCR(L1) | 6.13 | 0.850 |
| | (+0.30) | (+0.011) | | (+0.23) | (+0.002) |

## 4.5 GENERALIZATION TO UNSEEN TARGETS

In real-world regression tasks, it is common that some targets are unseen during training. As in Yang et al. (2021), we curate two subset of IMDB-WIKI that contain unseen targets. The test set is a uniform distribution across the whole target range. Table 6 shows the label distributions of the two training set, where pink shading indicates regions of unseen targets, and the blue shading represents the distribution of seen targets. The first training set consists of 3,781 training samples with a bi-modal Gaussian dis-

Table 6: Generalization performance to unseen targets on two curated subsets of IMDB-WIKI. MAE$^\downarrow$ is used as the metric.

| Label Distribution | Method | All | Seen | Unseen |
|---|---|---|---|---|
| | L1 | 12.53 | 10.82 | 18.40 |
| | SUPCR(L1) | 11.69 | 10.46 | 15.92 |
| | | (+0.84) | (+0.36) | (+2.48) |
| | L1 | 11.94 | 10.43 | 14.98 |
| | SUPCR(L1) | 10.88 | 9.78 | 13.08 |
| | | (+1.06) | (+0.64) | (+1.90) |

tribution over the target space, and the second training set consists of 3,662 training samples with a tri-modal Gaussian distribution over the target space. We report the prediction error on the seen and unseen targets separately. The results show that SUPCR (L1) outperforms L1 by a larger margin on the unseen targets without sacrificing the performance on the seen targets.

## 4.6 DESIGN VARIANTS

**Similarity Measure and Projection Head.** We explore potential variants of our method to pick the best design. Table 7 compares the performance of our method ($\mathcal{L}_{\text{SupCR}}$ with similarity measure using the negative L2 norm) with alternative designs that use a different similarity measure and potentially support a projection head. The results are on the AgeDB dataset and using SUPCR(L1).

Specifically, the table shows the performance of using different feature similarity measures $\text{sim}(\cdot, \cdot)$. We see that among cosine similarity, i.e., $\text{sim}(\boldsymbol{v}_i, \boldsymbol{v}_j) = \frac{\boldsymbol{v}_i \cdot \boldsymbol{v}_j}{\|\boldsymbol{v}_i\|_2 \|\boldsymbol{v}_j\|_2}$, negative L1 norm, i.e.,

Table 7: Comparison to potential variants of $\mathcal{L}_{\text{SupCR}}$ and their performance on AgeDB.

| Loss Term | Similarity Measure | Projection Head | MAE$^{\downarrow}$ | $R^{2\uparrow}$ |
|---|---|---|---|---|
| $\mathcal{L}_{\text{SimCLR}}$ | cosine | ✓ | 9.59 | 0.666 |
| $\mathcal{L}_{\text{SimCLR}}$ | cosine | ✗ | 10.67 | 0.600 |
| $\mathcal{L}_{\text{SupCon}}$ | cosine | ✓ | 8.13 | 0.753 |
| $\mathcal{L}_{\text{SupCon}}$ | cosine | ✗ | 7.89 | 0.760 |
| $\mathcal{L}_{\text{SupCR}}$ | cosine | ✓ | 7.34 | 0.793 |
| $\mathcal{L}_{\text{SupCR}}$ | cosine | ✗ | 6.51 | 0.836 |
| $\mathcal{L}_{\text{SupCR}}$ | negative L1 norm | ✗ | 6.25 | 0.842 |
| $\mathcal{L}_{\text{SupCR}}$ | negative L2 norm | ✗ | **6.14** | **0.850** |

$\text{sim}(\boldsymbol{v}_i, \boldsymbol{v}_j) = -\|\boldsymbol{v}_i - \boldsymbol{v}_j\|_1$ and negative L2 norm, i.e., $\text{sim}(\boldsymbol{v}_i, \boldsymbol{v}_j) = -\|\boldsymbol{v}_i - \boldsymbol{v}_j\|_2$, negative L2 norm delivers the best performance.

The table also shows the performance with and without a projection head. To train the encoder network $f(\cdot) : X \to \mathbb{R}^{d_e}$, one can involve a projection head $h(\cdot) : \mathbb{R}^{d_e} \to \mathbb{R}^{d_p}$ during training to calculate the loss on $\{h(f(\tilde{x}_i))\}_{i=1 \in [2N]}$, and then discard the projection head to use $f(x_i)$ for inference. One can also directly calculate the loss on $\{f(\tilde{x}_i)\}_{i \in [2N]}$ and then use $f(x_i)$ for inference. SIMCLR (Chen et al., 2020a) and SUPCON (Khosla et al., 2020) both use the former approach.

For $\mathcal{L}_{\text{SimCLR}}$, the results show that adding the projection head can benefit the regression performance. This is because $\mathcal{L}_{\text{SimCLR}}$ aims to extract features that are invariant to data augmentations and can remove information that may be useful for the downstream regression task. For $\mathcal{L}_{\text{SupCon}}$ and $\mathcal{L}_{\text{SupCR}}$, however, it is better to train the encoder without the projection head, since both $\mathcal{L}_{\text{SupCon}}$ and $\mathcal{L}_{\text{SupCR}}$ leverage the label information to extract features that directly target on the downstream task.

Moreover, we see that using $\mathcal{L}_{\text{SupCon}}$ or $\mathcal{L}_{\text{SupCR}}$ as the loss function delivers better performance than using $\mathcal{L}_{\text{SimCLR}}$, verifying that it is helpful to utilize label information during encoder training. We also see that $\mathcal{L}_{\text{SupCR}}$ outperforms $\mathcal{L}_{\text{SupCon}}$ by a large margin, highlighting the superiority of $\mathcal{L}_{\text{SupCR}}$ which explicitly leverages the ordered relationships between samples for regression problems.

**Training Scheme.** There are usually three schemes to train the feature encoder: (1) *Linear probing*: first trains the feature encoder using the representation learning loss, then freezes the encoder and trains a linear regressor on top of it using a regression loss. (2) *Fine-tuning*: first trains

Table 8: Performance of alternative training schemes for the feature encoder on AgeDB.

| Scheme | Linear Probing | Fine-tuning | Regularization | L1 |
|---|---|---|---|---|
| MAE$^{\downarrow}$ | **6.14** | 6.36 | 6.42 | 6.63 |
| $R^{2\uparrow}$ | **0.850** | 0.844 | 0.833 | 0.828 |

the feature encoder using the representation learning loss, then fine-tunes the whole model using a regression loss. (3) *Regularization*: trains the whole model while jointly optimizing the representation learning loss and the regression loss.

Table 8 shows the performance on AgeDB for the three schemes using $\mathcal{L}_{\text{SupCR}}$ as the representation learning loss and L1 as the regression loss. We see that all of the three schemes can improve performance over using the regression loss L1 alone to train a model. Further, unlike classification problems where fine-tuning often delivers the best performance, freezing the feature encoder performs the best. This is because in the case of regression, back-propagating the L1 loss to the representation can destroy the order in the embedding space learned by $\mathcal{L}_{\text{SupCR}}$, which leads to poorer performance.

## 5 CONCLUSION

In this paper, we propose Supervised Contrastive Regression (SupCR), a framework that learns a regression-aware representation by contrasting samples against each other according to their target distance. The proposed framework is orthogonal to existing regression models, and can be used in combination with such models to improve performance. It achieves the state-of-the-art performance and consistently improves prior regression baselines across different datasets, tasks, and input modalities. It also improves the robustness to data corruptions, resilience to reduced training data, performance on transfer learning, and generalization to unseen targets.

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

# A  PROOFS

**Theorem 1** (Lower bound of $\mathcal{L}_{\text{SupCR}}$). *$L^\star$ is a lower bound of $\mathcal{L}_{\text{SupCR}}$, i.e., $\mathcal{L}_{\text{SupCR}} > L^\star$.*

*Proof.*

$$
\begin{aligned}
\mathcal{L}_{\text{SupCR}} &= -\frac{1}{2N(2N-1)} \sum_{i=1}^{2N} \sum_{j \in [2N] \setminus \{i\}} \log \frac{\exp(s_{i,j})}{\sum\limits_{k \in [2N] \setminus \{i\}, \, d_{i,k} \geq d_{i,j}} \exp(s_{i,k})} \\
&= -\frac{1}{2N(2N-1)} \sum_{i=1}^{2N} \sum_{m=1}^{M_i} \sum_{j \in [2N] \setminus \{i\}, \, d_{i,j} = D_{i,m}} \log \frac{\exp(s_{i,j})}{\sum\limits_{k \in [2N] \setminus \{i\}, \, d_{i,k} \geq D_{i,m}} \exp(s_{i,k})} \\
&= -\frac{1}{2N(2N-1)} \sum_{i=1}^{2N} \sum_{m=1}^{M_i} \sum_{j \in [2N] \setminus \{i\}, \, d_{i,j} = D_{i,m}} \log \frac{\exp(s_{i,j})}{\sum\limits_{k \in [2N] \setminus \{i\}, \, d_{i,k} = D_{i,m}} \exp(s_{i,k})} \\
&\quad + \frac{1}{2N(2N-1)} \sum_{i=1}^{2N} \sum_{m=1}^{M_i} \sum_{j \in [2N] \setminus \{i\}, \, d_{i,j} = D_{i,m}} \log \left( 1 + \frac{\sum\limits_{k \in [2N] \setminus \{i\}, \, d_{i,k} > D_{i,m}} \exp(s_{i,k} - s_{i,j})}{\sum\limits_{k \in [2N] \setminus \{i\}, \, d_{i,k} = D_{i,m}} \exp(s_{i,k} - s_{i,j})} \right) \\
&> -\frac{1}{2N(2N-1)} \sum_{i=1}^{2N} \sum_{m=1}^{M_i} \sum_{j \in [2N] \setminus \{i\}, \, d_{i,j} = D_{i,m}} \log \frac{\exp(s_{i,j})}{\sum\limits_{k \in [2N] \setminus \{i\}, \, d_{i,k} = D_{i,m}} \exp(s_{i,k})}.
\end{aligned}
\tag{2}
$$

$\forall i \in [2N], m \in [M_i]$, from Jensen's Inequality we have

$$
-\sum_{j \in [2N] \setminus \{i\}, \, d_{i,j} = D_{i,m}} \log \frac{\exp(s_{i,j})}{\sum\limits_{k \in [2N] \setminus \{i\}, \, d_{i,k} = D_{i,m}} \exp(s_{i,k})}
\tag{3}
$$
$$
\geq -n_{i,m} \log \left( \frac{1}{n_{i,m}} \sum_{j \in [2N] \setminus \{i\}, \, d_{i,j} = D_{i,m}} \frac{\exp(s_{i,j})}{\sum\limits_{k \in [2N] \setminus \{i\}, \, d_{i,k} = D_{i,m}} \exp(s_{i,k})} \right) = n_{i,m} \log n_{i,m}.
$$

Thus, by plugging Equation 3 into Equation 2, we have

$$
\mathcal{L}_{\text{SupCR}} > \frac{1}{2N(2N-1)} \sum_{i=1}^{2N} \sum_{m=1}^{M_i} n_{i,m} \log n_{i,m} = L^\star.
\tag{4}
$$

$\square$

**Theorem 2** (Lower bound tightness). *For any $\epsilon > 0$, there exists a set of feature embeddings such that $\mathcal{L}_{\text{SupCR}} < L^\star + \epsilon$.*

*Proof.* We will show $\forall \epsilon > 0$, there is a set of feature embeddings where

$$
\begin{cases}
s_{i,j} > s_{i,k} + \gamma & \text{if } d_{i,j} < d_{i,k} \\
s_{i,j} = s_{i,k} & \text{if } d_{i,j} = d_{i,k}
\end{cases}
$$

and $\gamma := \log \frac{2N}{\min\limits_{i \in [2N], m \in [M_i]} n_{i,m} \epsilon}$, $\forall i \in [2N], j, k \in [2N] \setminus \{i\}$, such that $\mathcal{L}_{\text{SupCR}} < L^\star + \epsilon$.

For such a set of feature embeddings, $\forall i \in [2N], m \in [M_i], j \in \{j \in [2N] \setminus \{i\} \mid d_{i,j} = D_{i,m}\}$,

$$
-\log \frac{\exp(s_{i,j})}{\sum\limits_{k \in [2N] \setminus \{i\}, \, d_{i,k} = D_{i,m}} \exp(s_{i,k})} = \log n_{i,m}
\tag{5}
$$

since $s_{i,k} = s_{i,j}$ for all $k$ such that $d_{i,k} = D_{i,m} = d_{i,j}$, and

$$\log\left(1 + \frac{\sum\limits_{k\in[2N]\setminus\{i\},\,d_{i,k}>D_{i,m}} \exp(s_{i,k}-s_{i,j})}{\sum\limits_{k\in[2N]\setminus\{i\},\,d_{i,k}=D_{i,m}} \exp(s_{i,k}-s_{i,j})}\right) \tag{6}$$

$$< \log\left(1 + \frac{2N\exp(-\gamma)}{n_{i,m}}\right) < \frac{2N\exp(-\gamma)}{n_{i,m}} \leq \epsilon$$

since $s_{i,k} - s_{i,j} < -\gamma$ for all $k$ such that $d_{i,k} > D_{i,m} = d_{i,j}$ and $s_{i,k} - s_{i,j} = 0$ for all $k$ such that $d_{i,k} = D_{i,m} = d_{i,j}$.

From Equation 2 we have

$$\mathcal{L}_{\text{SupCR}} = -\frac{1}{2N(2N-1)} \sum_{i=1}^{2N} \sum_{m=1}^{M_i} \sum_{j\in[2N]\setminus\{i\},\,d_{i,j}=D_{i,m}} \log \frac{\exp(s_{i,j})}{\sum\limits_{k\in[2N]\setminus\{i\},\,d_{i,k}=D_{i,m}} \exp(s_{i,k})}$$

$$+ \frac{1}{2N(2N-1)} \sum_{i=1}^{2N} \sum_{m=1}^{M_i} \sum_{j\in[2N]\setminus\{i\},\,d_{i,j}=D_{i,m}} \log\left(1 + \frac{\sum\limits_{k\in[2N]\setminus\{i\},\,d_{i,k}>D_{i,m}} \exp(s_{i,k}-s_{i,j})}{\sum\limits_{k\in[2N]\setminus\{i\},\,d_{i,k}=D_{i,m}} \exp(s_{i,k}-s_{i,j})}\right), \tag{7}$$

By plugging Equation 5 and 6 into Equation 7 we have

$$\mathcal{L}_{\text{SupCR}} < \frac{1}{2N(2N-1)} \sum_{i=1}^{2N} \sum_{m=1}^{M_i} n_{i,m} \log n_{i,m} + \epsilon = L^\star + \epsilon \tag{8}$$

$\square$

**Theorem 3** (Main theorem). *For any $0 < \delta < 1$, there exist $\epsilon > 0$, such that if $\mathcal{L}_{\text{SupCR}} < L^\star + \epsilon$, then the feature embeddings are $\delta$-ordered.*

*Proof.* We will show $\forall 0 < \delta < 1$, there is a

$$\epsilon = \frac{1}{2N(2N-1)} \min\left(\min_{i\in[2N],m\in[M_i]} \log\left(1 + \frac{1}{n_{i,m}\exp(\delta+\frac{1}{\delta})}\right), 2\log\frac{1+\exp(\delta)}{2} - \delta\right) > 0,$$

such that when $\mathcal{L}_{\text{SupCR}} < L^\star + \epsilon$, the feature embeddings are $\delta$-ordered.

We first show that $|s_{i,j} - s_{i,k}| < \delta$ if $d_{i,j} = d_{i,k}, \forall i \in [2N], j,k \in [2N]\setminus\{i\}$ when $\mathcal{L}_{\text{SupCR}} < L^\star + \epsilon$.

From Equation 2 we have

$$\mathcal{L}_{\text{SupCR}} > -\frac{1}{2N(2N-1)} \sum_{i=1}^{2N} \sum_{m=1}^{M_i} \sum_{j\in[2N]\setminus\{i\},\,d_{i,j}=D_{i,m}} \log \frac{\exp(s_{i,j})}{\sum\limits_{k\in[2N]\setminus\{i\},\,d_{i,k}=D_{i,m}} \exp(s_{i,k})}. \tag{9}$$

Let $p_{i,m} := \arg\min\limits_{j\in[2N]\setminus\{i\},\,d_{i,j}=D_{i,m}} s_{i,j}$ , $q_{i,m} := \arg\max\limits_{j\in[2N]\setminus\{i\},\,d_{i,j}=D_{i,m}} s_{i,j}$, $\zeta_{i,m} := s_{i,p_{i,m}}, \eta_{i,m} := s_{i,q_{i,m}} - s_{i,p_{i,m}}, \forall i \in [2N], m \in [M_i]$, by splitting out the maximum term and the minimum term we have

$$\mathcal{L}_{\text{SupCR}} > -\frac{1}{2N(2N-1)} \sum_{i=1}^{2N} \sum_{m=1}^{M_i} \left\{ \log \frac{\exp(\zeta_{i,m})}{\sum\limits_{k\in[2N]\setminus\{i\},\,d_{i,k}=D_{i,m}} \exp(s_{i,k})} \right.$$

$$\left. + \log \frac{\exp(\zeta_{i,m} + \eta_{i,m})}{\sum\limits_{k\in[2N]\setminus\{i\},\,d_{i,k}=D_{i,m}} \exp(s_{i,k})} + \log \frac{\exp\left(\sum\limits_{j\in[2N]\setminus\{i,p_{i,m},q_{i,m}\},\,d_{i,j}=D_{i,m}} s_{i,j}\right)}{\left(\sum\limits_{k\in[2N]\setminus\{i\},\,d_{i,k}=D_{i,m}} \exp(s_{i,k})\right)^{n_{i,m}-2}} \right\}. \tag{10}$$

Let $\theta_{i,m} := \frac{1}{n_{i,m}-2} \sum_{j \in [2N] \setminus \{i, p_{i,m}, q_{i,m}\}, d_{i,j}=D_{i,m}} \exp(s_{i,j} - \zeta_{i,m})$, we have

$$- \log \frac{\exp(\zeta_{i,m})}{\sum_{k \in [2N] \setminus \{i\},\, d_{i,k}=D_{i,m}} \exp(s_{i,k})} = \log(1 + \exp(\eta_{i,m}) + (n_{i,m} - 2)\theta_{i,m}) \tag{11}$$

and

$$- \log \frac{\exp(\zeta_{i,m} + \eta_{i,m})}{\sum_{k \in [2N] \setminus \{i\},\, d_{i,k}=D_{i,m}} \exp(s_{i,k})} = \log(1 + \exp(\eta_{i,m}) + (n_{i,m} - 2)\theta_{i,m}) - \eta_{i,m}. \tag{12}$$

Then, from Jensen's inequality we know

$$\exp\left( \sum_{j \in [2N] \setminus \{i, p_{i,m}, q_{i,m}\}, d_{i,j}=D_{i,m}} s_{i,j} \right) \leq \left( \frac{1}{n_{i,m}-2} \sum_{j \in [2N] \setminus \{i, p_{i,m}, q_{i,m}\}, d_{i,j}=D_{i,m}} \exp(s_{i,j}) \right)^{n_{i,m}-2}, \tag{13}$$

thus

$$- \log \frac{\exp\left( \sum_{j \in [2N] \setminus \{i, p_{i,m}, q_{i,m}\}, d_{i,j}=D_{i,m}} s_{i,j} \right)}{\left( \sum_{k \in [2N] \setminus \{i\},\, d_{i,k}=D_{i,m}} \exp(s_{i,k}) \right)^{n_{i,m}-2}} \geq (n_{i,m}-2)\log(1 + \exp(\eta_{i,m}) + (n_{i,m}-2)\theta_{i,m}) - (n_{i,m}-2)\log(\theta_{i,m}) \tag{14}$$

By plugging Equation 11, 12 and 14 into Equation 10, we have

$$\mathcal{L}_{\text{SupCR}} > \frac{1}{2N(2N-1)} \sum_{i=1}^{2N} \sum_{m=1}^{M_i} \left( n_{i,m} \log(1 + \exp(\eta_{i,m}) + (n_{i,m}-2)\theta_{i,m}) - \eta_{i,m} - (n_{i,m}-2)\log(\theta_{i,m}) \right). \tag{15}$$

Let $h(\theta) := n_{i,m} \log(1 + \exp(\eta_{i,m}) + (n_{i,m}-2)\theta) - \eta_{i,m} - (n_{i,m}-2)\log(\theta)$. From derivative analysis we know $h(\theta)$ decreases monotonically when $\theta \in \left[ 1, \frac{1+\exp(\eta_{i,m})}{2} \right]$ and increases monotonically when $\theta \in \left[ \frac{1+\exp(\eta_{i,m})}{2}, \exp(\eta_{i,m}) \right]$, thus

$$h(\theta) \geq h\left( \frac{1+\exp(\eta_{i,m})}{2} \right) = n_{i,m} \log n_{i,m} + 2\log \frac{1+\exp(\eta_{i,m})}{2} - \eta_{i,m}. \tag{16}$$

By plugging Equation 16 into Equation 15, we have

$$\mathcal{L}_{\text{SupCR}} > \frac{1}{2N(2N-1)} \sum_{i=1}^{2N} \sum_{m=1}^{M_i} \left( n_{i,m} \log n_{i,m} + 2\log \frac{1+\exp(\eta_{i,m})}{2} - \eta_{i,m} \right)$$

$$= L^\star + \frac{1}{2N(2N-1)} \sum_{i=1}^{2N} \sum_{m=1}^{M_i} \left( 2\log \frac{1+\exp(\eta_{i,m})}{2} - \eta_{i,m} \right). \tag{17}$$

Then, since $\eta_{i,m} \geq 0$, we have $2\log \frac{1+\exp(\eta_{i,m})}{2} - \eta_{i,m} \geq 0$. Thus, $\forall i \in [2N], m \in [M_i]$,

$$\mathcal{L}_{\text{SupCR}} > L^\star + \frac{1}{2N(2N-1)} \left( 2\log \frac{1+\exp(\eta_{i,m})}{2} - \eta_{i,m} \right). \tag{18}$$

If $\mathcal{L}_{\text{SupCR}} < L^\star + \epsilon \leq L^\star + \frac{1}{2N(2N-1)} \left( 2\log \frac{1+\exp(\delta)}{2} - \delta \right)$, then

$$2\log \frac{1+\exp(\eta_{i,m})}{2} - \eta_{i,m} < 2\log \frac{1+\exp(\delta)}{2} - \delta. \tag{19}$$

Since $y(x) = 2\log \frac{1+\exp(x)}{2} - x$ increases monotonically when $x > 0$, we have $\eta_{i,m} < \delta$. Hence, $\forall i \in [2N], j, k \in [2N] \setminus \{i\}$, if $d_{i,j} = d_{i,k} = D_{i,m}, |s_{i,j} - s_{i,k}| \leq \eta_{i,m} < \delta$.

Next, we show $s_{i,j} > s_{i,k} + \delta$ if $d_{i,j} < d_{i,k}$ when $\mathcal{L}_{\text{SupCR}} < L^\star + \epsilon$.

From Equation 2 we have

$$\mathcal{L}_{\text{SupCR}} = -\frac{1}{2N(2N-1)} \sum_{i=1}^{2N} \sum_{m=1}^{M_i} \sum_{j\in[2N]\backslash\{i\},\, d_{i,j}=D_{i,m}} \log \frac{\exp(s_{i,j})}{\sum_{k\in[2N]\backslash\{i\},\, d_{i,k}=D_{i,m}} \exp(s_{i,k})}$$
$$+ \frac{1}{2N(2N-1)} \sum_{i=1}^{2N} \sum_{m=1}^{M_i} \sum_{j\in[2N]\backslash\{i\},\, d_{i,j}=D_{i,m}} \log \left(1 + \frac{\sum_{k\in[2N]\backslash\{i\},\, d_{i,k}>D_{i,m}} \exp(s_{i,k}-s_{i,j})}{\sum_{k\in[2N]\backslash\{i\},\, d_{i,k}=D_{i,m}} \exp(s_{i,k}-s_{i,j})}\right),$$
(20)

and combining it with Equation 3 we have

$$\mathcal{L}_{\text{SupCR}} \geq L^\star + \frac{1}{2N(2N-1)} \sum_{i=1}^{2N} \sum_{m=1}^{M_i} \sum_{j\in[2N]\backslash\{i\},\, d_{i,j}=D_{i,m}} \log \left(1 + \frac{\sum_{k\in[2N]\backslash\{i\},\, d_{i,k}>D_{i,m}} \exp(s_{i,k}-s_{i,j})}{\sum_{k\in[2N]\backslash\{i\},\, d_{i,k}=D_{i,m}} \exp(s_{i,k}-s_{i,j})}\right)$$
$$> L^\star + \frac{1}{2N(2N-1)} \log \left(1 + \frac{\exp(s_{i,k}-s_{i,j})}{\sum_{l\in[2N]\backslash\{i\},\, d_{i,l}=d_{i,j}} \exp(s_{i,l}-s_{i,j})}\right),$$
(21)

$\forall i \in [2N]$, $j \in [2N]\backslash\{i\}, k \in \{k \in [2N]\backslash\{i\} \mid d_{i,j} < d_{i,k}\}$.

When $\mathcal{L}_{\text{SupCR}} < L^\star + \epsilon$, we already have $|s_{i,l} - s_{i,j}| < \delta, \forall d_{i,l} = d_{i,j}$, which derives $s_{i,l} - s_{i,j} < \delta$ and thus $\exp(s_{i,l} - s_{i,j}) < \exp(\delta)$. By putting this into Equation 20, we have $\forall i \in [2N]$, $j \in [2N]\backslash\{i\}, k \in \{k \in [2N]\backslash\{i\} \mid d_{i,j} < d_{i,k}\}$,

$$\mathcal{L}_{\text{SupCR}} > L^\star + \frac{1}{2N(2N-1)} \log \left(1 + \frac{\exp(s_{i,k}-s_{i,j})}{n_{i,r_{i,j}} \exp(\delta)}\right),$$
(22)

where $r_{i,j} \in [M_i]$ is the index such that $D_{i,r_{i,j}} = d_{i,j}$.

Further, given $\mathcal{L}_{\text{SupCR}} < L^\star + \epsilon < L^\star + \frac{1}{2N(2N-1)} \log(1 + \frac{1}{n_{i,r_{i,j}} \exp(\delta + \frac{1}{\delta})})$, we have

$$\log(1 + \frac{\exp(s_{i,k}-s_{i,j})}{n_{i,r_{i,j}} \exp(\delta)}) < \log(1 + \frac{1}{n_{i,r_{i,j}} \exp(\delta + \frac{1}{\delta})})$$
(23)

which derives $s_{i,j} > s_{i,k} + \frac{1}{\delta}, \forall i \in [2N]$, $j \in [2N]\backslash\{i\}, k \in \{k \in [2N]\backslash\{i\} \mid d_{i,j} < d_{i,k}\}$.

Finally, $\forall i \in [2N]$, $j, k \in [2N]\backslash\{i\}$, $s_{i,j} < s_{i,k} - \frac{1}{\delta}$ if $d_{i,j} > d_{i,k}$ directly follows from $s_{i,j} > s_{i,k} + \frac{1}{\delta}$ if $d_{i,j} < d_{i,k}$.

□

# B  ETHICS STATEMENTS

All of the datasets used in the paper are public datasets. The ethics statements for each dataset are listed below:

- TUAB: This dataset is collected from archival records at Temple University Hospital (TUH). All work was performed in accordance with the Declaration of Helsinki and with the full approval of the Temple University IRB. All personnel in contact with privileged patient information were fully trained on patient privacy and were certified by the Temple IRB (Obeid & Picone, 2016).
- MPIIFaceGaze: The authors of Zhang et al. (2017b) confirmed that the informed consent has been obtained from all of the participants.
- AgeDB, IMDB-WIKI and SkyFinder: Those datasets are collected without including any interaction or intervention with human subjects and do not contain any private information (Moschoglou et al., 2017; Rothe et al., 2015; Mihail et al., 2016).

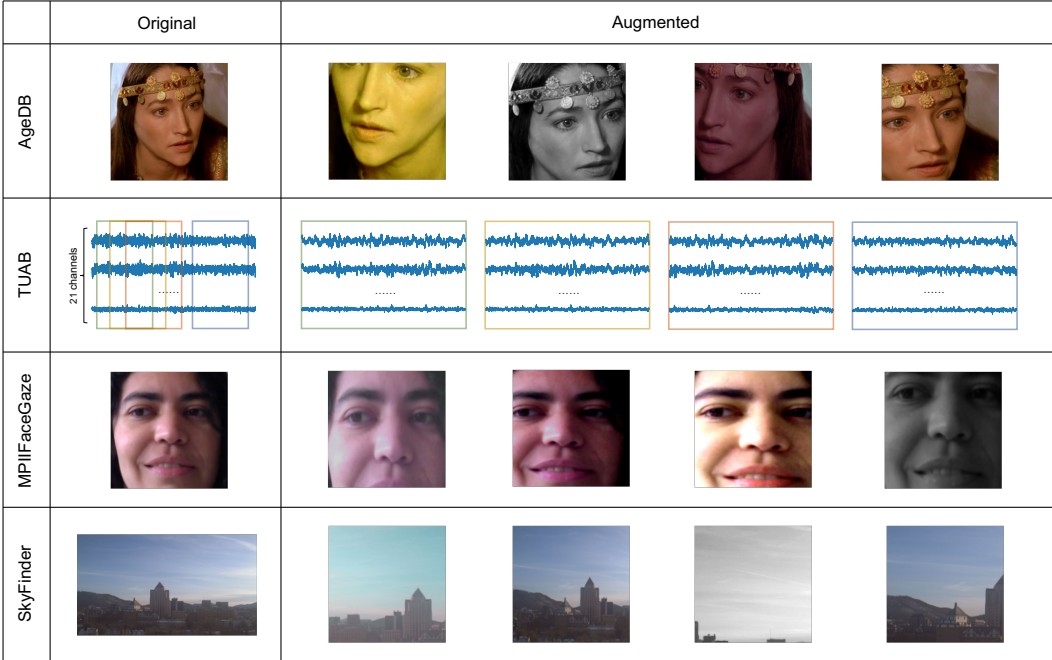

Figure 5: Visualizations of original and augmented data samples in each dataset.

## C  VISUALIZATION OF DATA AUGMENTATION

Figure 5 shows examples of original and augmented data samples on each dataset. The data augmentation used on each dataset are listed below:

- For `AgeDB` and `SkyFinder`, random crop and resize (with random horizontal flip), color distortions are used as data augmentation;

- For `TUAB`, random crop is used as data augmentation;

- For `MPIIFaceGaze`, random crop and resize (without random horizontal flip), color distortions are used as data augmentation.

## D  DETAILED EXPERIMENT SETTINGS

For the encoder training of our method and regression learning baselines, we selected the best learning rates and weight decays for each dataset by grid search, with a grid of learning rates from $\{0.01, 0.05, 0.1, 0.2, 0.5, 1.0\}$ and weight decays from $\{10^{-6}, 10^{-5}, 10^{-4}, 10^{-3}\}$. For the predictor training of our method, we adopted the same search setting above except for adding no weight decay to the search choices of weight decays. For temperature parameter $\tau$, we searched from $\{0.1, 0.2, 0.5, 1.0, 2.0, 5.0\}$ and selected the best, which is 2.0.

For the classification-based baselines, we divided the regression range into small bins. For `AgeDB`, the target range is $0 \sim 101$ and the bin size is set to 1; for `TUAB`, the target range is $0 \sim 95$ and the bin size is set to 1; for `MPIIFaceGaze`, the target range is -40 $\sim$ 10 for the pitch angle and -45 $\sim$ 45 for the yaw angle, and the bin size is set to 0.5 for the pitch angle and is set to 1 for the yaw angle. For `SkyFinder`, the target range is -20 $\sim$ 49 and the bin size is set to 1. For the implementation of SupCon (Khosla et al., 2020) on regression datasets, the regression range is divided into small bins in the same way and each bin is regarded as a class. Samples from the same class, i.e., sharing the same target bin as the anchor, are considered as positive samples.

# E    ADDITIONAL EXPERIMENTS AND ANALYSIS

## E.1    IMPACT OF MODEL ARCHITECTURES

In the main paper, we use ResNet-18 as the encoder backbone for three image datasets (`AgeDB`, `MPIIFaceGaze` and `SkyFinder`). In this section, we study the impact of backbone architectures on the experiment results. As Table 9 reports, the results of using ResNet-50 as the encoder backbone are consistent with the results with ResNet-18 in Table 1, Table 3 and Table 4, showing that our method is compatiable with different model architectures.

Table 9: Evaluation results using ResNet-50 as the encoder backbone.

| Dataset | AgeDB | | MPIIFaceGaze | | SkyFinder | |
|---|---|---|---|---|---|---|
| Metrics | MAE$^\downarrow$ | $R^{2\uparrow}$ | Angular$^\downarrow$ | $R^{2\uparrow}$ | MAE$^\downarrow$ | $R^{2\uparrow}$ |
| L1 | 6.49 | 0.830 | 5.74 | 0.748 | 2.88 | 0.863 |
| SUPCR(L1) | **6.10** | **0.851** | **5.16** | **0.819** | **2.78** | **0.877** |
| GAINS | +0.39 | +0.021 | +0.58 | +0.071 | +0.10 | +0.014 |

## E.2    COMPARISON TO EXISTING SOTA METHODS ON EACH DATASET

In this section, we compare the performance of our method with the current state-of-the-art (SOTA) methods on each dataset. To the best of our knowledge, the SOTA on each dataset are (Yang et al., 2021) for `AgeDB`, (Engemann et al., 2022) for `TUAB`, (Abdelrahman et al., 2022) for `MPIIFaceGaze`, and (Chu et al., 2018) for `SkyFinder`. However, They are either using different backbone architectures and optimizers, or using different dataset splits and number of training samples from ours. Therefore, the results reported in their paper are not directly comparable to our results. We implement their method under our training and evaluation protocols and summarize the results in Table 10. Results show that our method surpasses the state-of-the-art methods on all datasets.

Table 10: Comparison to exsiting SOTA methods on each dataset.

| Dataset | AgeDB | TUAB | | MPIIFaceGaze | SkyFinder | |
|---|---|---|---|---|---|---|
| Metrics | MAE$^\downarrow$ | MAE$^\downarrow$ | $R^{2\uparrow}$ | Angular$^\downarrow$ | MAE$^\downarrow$ | RMSE$^\downarrow$ |
| SOTA | 6.58 | 7.96 | 0.599 | 5.76 | 2.97 | 4.05 |
| SUPCR | **6.14** | **6.91** | **0.697** | **5.13** | **2.85** | **3.84** |
| GAINS | +0.44 | +1.05 | +0.098 | +0.63 | +0.12 | +0.21 |

## E.3    COMPARISON TO MORE SELF-SUPERVISED LEARNING METHODS

In Section 4.6, we compare our method with SIMCLR (Chen et al., 2020a) and SUPCON (Khosla et al., 2020) on `AgeDB` dataset. In this section, we investigate more SSL methods under various training settings to compare with our method on all datasets. Specifically, we evaluate the encoder pretrained using SIMCLR and DINO (Caron et al., 2021) on ImageNet and the regression datasets respectively. Besides, we also consider the setting where we use SIMCLR loss as regularization along with L1 loss to train on the regression datasets. The results are shown in Table 11. We use ResNet-50 as the encoder backbone for image datasets. The ResNet-50 ImageNet-pretrained encoders are 800-epoch checkpoints and their ImageNet linear evaluation accuracies are 69.1% and 75.3% respectively. For DINO-related methods, apart from linear evaluation, we also report $k$-NN evaluation results ($k = 20$).

The results show that SUPCR outperforms all these baselines on all datasets, and pre-training using SIMCLR or DINO, or using SIMCLR as a regularizer all performs worse than the vanilla L1 baseline. This further verifies that the performance gain of our method stems from our proposed SUPCR loss rather than the pre-training scheme.

Table 11: Comparison to SSL methods applied under different settings on each dataset. MAE$^\downarrow$ is used as the metric for `AgeDB`, `TUAB` and `SkyFinder`, and Angular Error$^\downarrow$ for `MPIIFaceGaze`.

| Dataset | AgeDB | | TUAB | | MPIIFaceGaze | | SkyFinder | |
|---|---|---|---|---|---|---|---|---|
| Evaluation Protocol (for pre-training scheme) | Linear | $k$-NN | Linear | $k$-NN | Linear | $k$-NN | Linear | $k$-NN |
| L1 | 6.49 | | 7.96 | | 5.74 | | 2.88 | |
| L1 + SIMCLR REGULARIZATION | 6.53 | | 8.02 | | 5.85 | | 2.93 | |
| SIMCLR (PRE-TRAINED ON IMAGENET) | 15.32 | – | – | | 16.11 | – | 5.03 | – |
| DINO (PRE-TRAINED ON IMAGENET) | 9.14 | 11.82 | – | | 12.44 | 12.84 | 4.62 | 3.64 |
| SIMCLR (PRE-TRAINED ON REGRESSION DATASET) | 9.38 | – | 11.01 | – | 9.13 | – | 4.55 | – |
| DINO (PRE-TRAINED ON REGRESSION DATASET) | 9.91 | 12.03 | 11.62 | 12.32 | 11.90 | 12.50 | 5.00 | 4.20 |
| SUPCR(L1) | **6.10** | – | **6.97** | – | **5.16** | – | **2.78** | – |

### E.4 STANDARD DEVIATIONS OF RESULTS

In this section, we study the standard deviations of the best results on each dataset with 5 different random seeds. Table 12 shows their average prediction errors and standard deviations. These results are aligned with the results we reported in the main paper.

Table 12: Average prediction errors and standard deviations of the best results on each dataset.

| AgeDB: SUPCR(L1) | TUAB: SUPCR(DLDL-V2) | MPIIFaceGaze: SUPCR(OR) | SkyFinder: SUPCR(DLDL-V2) |
|---|---|---|---|
| $6.19 \pm 0.08$ | $7.00 \pm 0.10$ | $5.24 \pm 0.13$ | $2.87 \pm 0.04$ |

### E.5 ABLATION ON NUMBER OF POSITIVES

In $\mathcal{L}_{\text{SupCR}}$, all samples in the batch will be treated as the positive for each anchor. Here, we conduct an ablation study on only considering the first $K$ closest samples to the anchor as positive. Table 13 shows the MAE on `AgeDB` for different $K$s (using L1 loss in the linear probing), where $K = 511$ means considering all samples as positive since we use a batch size $N = 256$ and thus the number of all samples other than the anchor equal to $2N - 1 = 511$. These experiments show that the larger $K$, the better the performance.

Table 13: Ablation on number of positives considered.

| $K$ | 128 | 256 | 384 | 511 |
|---|---|---|---|---|
| MAE | 6.46 | 6.43 | 6.29 | 6.14 |

This phenomenon is aligned with the design intuition of SUPCR loss since each contrastive term ensures a group of orders related to the positive sample, i.e., it ensures all samples that have a larger label distance from the anchor than the positive sample to be farther from the anchor than the positive in the feature embedding space. Only when all of the samples are considered as positive and their related groups of orders are ensured, the order in the feature space can be fully guaranteed.

### E.6 TRAINING EFFICIENCY

We compute the average wall-clock running time (in seconds) per training epoch on 8 NVIDIA TITAN RTX GPUs for SUPCR and compare it with SUPCON (Khosla et al., 2020) on all four datasets, as shown in Table 14. Results show the training efficiency of SUPCR is comparable to SUPCON.

Table 14: Average wall-clock running time (in seconds) per training epoch on 8 NVIDIA TITAN RTX GPUs for SUPCR and SUPCON (Khosla et al., 2020) on each dataset.

| Dataset | AgeDB | TUAB | MPIIFaceGaze | SkyFinder |
|---|---|---|---|---|
| SUPCON | 23.1 | 25.3 | 69.1 | 55.6 |
| SUPCR | 26.2 | 27.3 | 75.4 | 61.8 |
| RATIO | 1.13 | 1.08 | 1.09 | 1.11 |

