# OpenReview forum: "Supervised Contrastive Regression"
_ICLR.cc/2023/Conference — Submitted to ICLR 2023_

### Official Review · Reviewer_7jug · 2022-10-23

**Confidence:** 4
**Correctness:** 4
**Technical Novelty And Significance:** 2
**Empirical Novelty And Significance:** 3
**Recommendation:** 6

**Clarity, Quality, Novelty And Reproducibility:**

The paper is clearly written with extensive experiments. The originality of the work is limited as it is adapting SupCon approach for regression tasks.

**Strength And Weaknesses:**

Strengths -
1. The paper is well written and easy to follow along.
2. The authors do an extensive study across different approaches and settings, such as which similarity function to use, whether to use projection head or now and others.
3. Along with better task performance, Improved robustness to corruption and generalization to unseen targets shows the goodness of the approach.

Weakness -

1. For the contrastive learning part, the paper iteratively considers the first closest sample, then the second closest and so on for their positive samples. It is unclear if we really need to consider this iteration till 2N steps. What happens if we only consider the first closest of the first K closest samples as positive. This ablation study would be important to justify the proposed approach, but is missing from the paper.

2. It's unclear how is SupCon used as a baseline in the paper, when the task is a regression task. How are the authors constructing positive samples for the SupCon approach?



**Summary Of The Paper:**

This paper proposes a new method for regression problems based on Supervised Contrastive Learning approach(SupCon) designed for classification tasks. The paper argues that existing regression-based approaches, don't learn embeddings suited for regression tasks, whereby the distance in embedding space is equivalent to the distance in the label space. They empirically show this claim by a UMAP plot for a regression task, whereby the embeddings learned by baseline L1 approaches do not capture the intrinsic ordered relationship between the samples. The paper argues that learning a regression-suited embedding space, will allow the models to be robust to corruption and also generalize to unseen targets.
Thus, the authors propose an approach similar to SupCon for regression tasks which they call SupCR(Supervised Contrastive Regression). In SupCon the positive samples for contrastive learning are not only the augmented version of the query but also those that belong to the same class as the query. In SupCR, the authors first consider the closest sample as the positive, and all other samples whose label distance is larger than the distance between the query and positive are treated as the negative. This is then iterated for the second closest, third closest samples and so on. Once the embeddings are learned in the contrastive way, the authors either fine-tune the entire network or only train a linear layer using either L1/MSE/Huber loss.

The authors show that their proposed approach achieves state-of-the-art results over several different datasets. Additionally, the proposed method shows robustness to corruption, better transfer learning abilities, and generalization to unseen targets.
The authors also provide a theoretical justification for how their proposed optimization will make the feature embeddings ordered according to order in the label space.

**Summary Of The Review:**

While the paper does extensive study and provides a good justification of the proposed approach via state-of-the-art results, the novelty of the work is a bit limited as it is mostly just adapting SupCon for regression tasks. I also have a few more concerns regarding the further justification of the proposed approach and the baselines.
I am leaning towards weak accept but am willing to update my ratings if my concerns are addressed.

---

> ### Author Response · Authors · 2022-11-16
> **Response to Reviewer 7jug**
>
> Dear Reviewer 7jug,
>
> We thank the reviewer for the constructive comments and insightful feedback, which are very helpful in improving the quality of our paper. Below, we provide additional results and clarifications to address the reviewer’s concerns one by one, which we hope will encourage the reviewer to further increase the score.
>
> > For the contrastive learning part, the paper iteratively considers the first closest sample, then the second closest and so on for their positive samples. It is unclear if we really need to consider this iteration till 2N steps. What happens if we only consider the first K closest samples as positive. This ablation study would be important to justify the proposed approach, but is missing from the paper.
>
> Thanks for raising this interesting point. We did experiment for different Ks on the AgeDB dataset (using L1 loss in the linear probing) when only considering the first K closest samples as positive and the results are shown as follows:
>
> | K   | 128  | 256  | 384  | 511  |
> |-----|------|------|------|------|
> | MAE | 6.46 | 6.43 | 6.29 | 6.14 |
>
> where k = 511 means considering all of the samples since we use a batch size N = 256 and thus the number of all samples other than the anchor equal to 2N - 1 = 511. We will add these experiments in the paper. These experiments show that the larger K, the better the performance.
>
> Further, we would like to emphasize that each contrastive term ensures a group of orders related to the positive sample, i.e., it ensures all samples that have a larger label distance from the anchor than the positive sample to be farther from the anchor than the positive in the feature embedding space. Only when all of the samples are considered as positive and their related groups of orders are ensured, the order in the feature space can be fully guaranteed.
>
> **We have added these experiments and discussion in Appendix E.5.**
>
> > It's unclear how is SupCon used as a baseline in the paper, when the task is a regression task. How are the authors constructing positive samples for the SupCon approach?
>
> Thanks for pointing this out and we apologize for the missing description about this. A regression task can be converted into a classification task by dividing the target range into small bins and each bin will be regarded as a class. This is widely-used in classification-based methods for regression problems [1, 2]. Accordingly, for SupCon, samples from the same class, i.e., sharing the same target bin as the anchor, are considered as positive samples. **We have added the details about this in Appendix D.**
>
> [1] Rothe et al. DEX: Deep EXpectation of Apparent Age From a Single Image. ICCV 2015.
>
> [2] Abdelrahman et al. L2CS-Net: Fine-Grained Gaze Estimation in Unconstrained Environments. Arxiv 2022.
>
> > The novelty of the work is a bit limited as it is mostly just adapting SupCon for regression tasks.
>
> We would like to clarify that our proposed method is **not** just simply adapting SupCon for regression tasks. There are some simple extensions of SupCon for regression tasks, such as converting regression to a classification problem to apply SupCon (as mentioned in the above question and for which the results on AgeDB are shown in Table 7), or reweighting the contrastive loss by label similarity [3] (as mentioned in the related works, it improves domain adaptation but provides no benefits on the source dataset). Such adaptations **cannot** provide satisfactory performance on regression problems since their designs fail to capture the intrinsic ordered relationships between samples for regression problems, and reflect it onto the representation space.
>
> In contrast, our proposed method manages to learn a regression-aware representation space by contrasting samples against each other based on their target distance and we provide a theoretical analysis to show that optimizing our $L_{SupCR}$ loss will make the feature embedding ordered according to the order in the label space. Empirical results verify that our proposed method consistently improves the performance of the regression learning baselines – i.e., our method can be composed with other regression algorithms to further improve their performance.
>
> Therefore, we believe our work is **novel**, which is also recognized by other reviewers, *“The quality and novelty of the work are high”* (Reviewer hEGc), *“The particular loss function proposed in the work is novel”* (Reviewer dbgX).
>
> [3] Wang et al 2022. Contrastive Regression for Domain Adaptation on Gaze Estimation. CVPR 2022.
>
> We appreciate hearing from the reviewer, and hope that our response has addressed the reviewer’s concerns.

---

> ### Author Response · Authors · 2022-11-29
> **We would like to hear back from Reviewer 7jug**
>
> Dear Reviewer 7jug,
>
> Thank you for your time and effort in reviewing our work. We believe we have addressed all your concerns and questions. We are happy to discuss and provide further clarifications. Considering that the discussion deadline is approaching, could you please kindly consider updating the reviews and raising your score in light of our response?
>
> Thanks,
>
> Authors

---

> > ### Author Response · Authors · 2022-12-11
> > **Discussion period ending soon; We would like to hear back from Reviewer 7jug**
> >
> > We would like to thank the reviewer again for your time and effort. We have provided additional experiments, results, and clarifications to respond to your comments. We believe these should have successfully addressed all your concerns.
> >
> > Given there are only ***2 days left*** in the discussion period, we wanted to check if the reviewer had seen our responses and whether there were any more clarifications the reviewer would like. If the concerns of the reviewer are clarified and the reviewer is convinced of the novelty, significance, and completeness of our work, we'd be grateful if the reviewer could update your review and increase your score to reflect that, so we know that our response has been seen.
> >
> > Thanks, Authors

---

### Official Review · Reviewer_dbgX · 2022-10-25

**Confidence:** 4
**Correctness:** 4
**Technical Novelty And Significance:** 2
**Empirical Novelty And Significance:** 3
**Recommendation:** 6

**Clarity, Quality, Novelty And Reproducibility:**

**Clarity/Quality:**

- The paper is clearly written.
- Experimental evaluation is sound, results are laid out clearly.

**Novelty:**

- The particular loss function proposed in the work is novel, to the best of my knowledge.
- The idea of "supervised contrastive regression" and its effectiveness has been demonstrated before and is not novel (which is not a weakness in itself - just the presentation thereof is currently problematic). The related work and intro should be adapted/expanded in this regard. A few examples of papers I am aware of beyond the already cited ones, there might be more:
    - [Dufumier et al. (November 2021)](https://arxiv.org/pdf/2111.05643.pdf) discuss a supervised contrastive loss by re-weighting the loss terms based on a kernel function, which they originally proposed in [Dufumier et al., (September 2021)](https://link.springer.com/chapter/10.1007/978-3-030-87196-3_6) [(pre-print, June 2021)](https://arxiv.org/pdf/2106.08808.pdf).
    - [Schneider et al. (April 2022)](https://arxiv.org/abs/2204.00673) propose an alternative formulation of "supervised contrastive regression" by changing the *positive* distribution according to a similarity function applied to the data with continuous, discrete or both continuous/discrete labels (from comparing the Figure 1 in this paper and Figure 1 in Schneider et al., the resulting embeddings also seem to be qualitatively quite similar).
- In the light of this and the other works cited under Related Work already, the paper title is too broad. I find it necessary to specialize the title into something like "Supervised contrastive regression by ranking negative samples" or similar (i.e., name the particular contribution in the title). It should be made clear that the technical contribution of the paper lies in the specific choice of loss function rather than "Supervised Contrastive Regression" in general. In the intro, paragraphs 3 and 4 should also be updated accordingly, which currently incorrectly claim that this paper introduces the notion of contrastive learning with continuous labels.

**Reproducibility:**

- The experimental setup is clear. It would be good to clarify whether the authors will full open source the code for the experimental setup/evaluation.

**Strength And Weaknesses:**

**Strengths:**

- The method is simple to implement, well evaluated and the results are laid out clearly.
- The authors demonstrate improvements of their method both in terms of accuracy and robustness to image perturbations on a range of different evaluation tasks.
- The method is supported by theory. *Disclaimer: While the theoretical results are plausible, I did not check the proofs yet. I might update this line after verifying them.*

**Major Weaknesses:**

- The models used for the regression tasks are relatively small. Do the results still hold when a more powerful backbone architecture (e.g., a ResNet50, or even more recent architectures) are used? What was the motivation for using a ResNet18?
- The evaluation against the baselines is thorough, but I am missing a comparison to conceptually different methods. Even if the proposed method does not outperform them, it is good to provide this as a reference. Can the authors comment on the SOTA models for each of the five benchmarks? E.g. for Table 3, I find a reference result of [around 3-5 angular error](https://paperswithcode.com/sota/gaze-estimation-on-mpii-gaze), while authors report >5. I would be interested in learning about the state of the art of each of the benchmarks (I am not experienced with these datasets, so maybe the tables already contain the applicable SOTA).
- Since all results are for visual data and the method uses augmentations for training the embedding, I am missing a comparison to state of the art representation learning methods like DINO (Caron et al., 2021) or also older methods such as SimCLR (Chen et al., 2020). Depending on the results, these do not necessarily need to go into the main paper, but it would be good to confirm that the boost in performance can actually be attributed to the supervised contrastive learning scheme, rather than the augmentations. The comparison could be executed within the proposed evaluation protocol (train the model on the data using the respective augmentations, then evaluate - this has been partially done in Table 7 for one of the datasets) and beyond it using the "extra" ImageNet-pretraining data (use the pre-trained checkpoint, evaluate the feature space). I am happy to further discuss the specifics of this experiment before the authors execute it to ensure that the protocol makes sense.
- The positioning in the literature is not accurate, and the paper title is too broad. The authors already cite two papers (and I propose three additional ones below) that perform contrastive learning on continuous labels for the purpose of regresson tasks. Please see below ("Novelty").
- The paper lacks an ethics statement given the use of data involving human subjects, and the nature of the tasks (e.g., age prediction). More details on the used datasets should be added. I did not flag the paper for ethics review yet because I assume that the original datasets included such a statement.

**Minor Weaknesses**

- Tables 1-4 (and also others) are lacking standard deviations / confidence intervals. For readers less familar with the datasets, it is hard to judge whether the performance improvement is signficant in all cases. If the authors want to save compute, at least running the best results for 3-5 seeds would be great.
- Details on the grid search are missing (search range, number of samples per hyperparameter, ...)

**Questions:**

- Are all datasets taken from an established benchmark, or is the particular experimental protocol used in the paper a novel contribution? What was the rationale for using these specific prediction tasks for evaluation?
- How did you find the optimal temperature parameter?

**Summary Of The Paper:**

The paper proposes a new variant performing supervised contrastive learning with continuous labels. Improvements on four image-based and one EEG regression tasks are demonstrated when using the learned feature space instead of training various baseline algorithms from scratch. The paper also shows improvements in terms of robustness and transfer learning performance, as well as ablations on the impact of a projection head on the regression performance.

**Summary Of The Review:**

The paper introduces a new loss function for supervised contrastive learning with continuous labels. The experimental setup is interesting and well evaluated, and the paper is well written. The main limitation is the insufficient discussion of prior work on contrastive learning with continuous labels and positioning in the literature, along with possible improvements in the experiments. I made specific suggestions on how to improve these weaknesses above, and am willing to adapt my scores.

---

## Post Rebuttal comments (for now)

The authors did a good job at addressing my concerns. I raised the *Correctness* from 3 to 4, especially because a lot of insightful baseline experiments have been added. I also appreciate that the authors now better position the work in the existing literature, and refined both the paper title and the discussion of prior work.

I raised my overall score from 5 to 6 (pending discussion with other reviewers).

---

> ### Author Response · Authors · 2022-11-09
> **Looking for your clarification**
>
> Dear Reviewer dbgX,
>
> Thanks for your constructive comments and insightful feedback. These are very helpful to further improve the quality of our paper. We are now in the process of writing the response to the reviews. Before posting the full response, we would like to confirm the detailed experiment settings for **the third point in the Major Weaknesses**.
>
> First we would like to clarify that *not all results* are for visual datasets (the input of TUAB dataset is EEG signals) and we apply the same augmentations to all methods (including our method, SSL methods, regression-learning baselines), as indicated in Section 4 Experiment Settings.
>
> As you have mentioned, in Table 7, we have shown the comparisons to SimCLR and SupCon on AgeDB dataset. The additional experiments we are aware of according to your suggestions are:
> - Use SimCLR and DINO to pretrain the encoder on regression datasets (except SimCLR on AgeDB) and then do linear probing evaluation;
> - Use the publicly available encoder pretrained on ImageNet using SimCLR and DINO respectively to do linear probing evaluation on regression datasets;
>
> We would like to confirm whether our understanding is correct or not. We also welcome any further clarifications or suggestions on the experiment settings above.
>
> Thanks,
>
> Authors

---

> > ### Comment · Reviewer_dbgX · 2022-11-10
> > **Re: Looking for your clarification**
> >
> > > First we would like to clarify that not all results are for visual datasets (the input of TUAB dataset is EEG signals) and we apply the same augmentations to all methods (including our method, SSL methods, regression-learning baselines), as indicated in Section 4 Experiment Settings.
> >
> > Acknowledged, I was aware of this and will update the review to make this clear. Sorry about the confusion.
> >
> > > Use SimCLR and DINO to pretrain the encoder on regression datasets (except SimCLR on AgeDB) and then do linear probing evaluation;
> > > Use the publicly available encoder pretrained on ImageNet using SimCLR and DINO respectively to do linear probing evaluation on regression datasets;
> >
> > Exactly. Since the discussion phase is limited, I would suggest to prioritize reporting of results in this order:
> >
> > 1. Setup experiments without retraining first, i.e. your second point as this is presumably very quick to do. For DINO you might want to test a kNN regressor as well, as the features are very well suited for kNN approaches on top.
> > 2. Retrain the models, same analysis as in (1).
> > 3. Evaluate (1) and (2) with all downstream regression algorithms you report in your result tables.

---

> ### Author Response · Authors · 2022-11-16
> **Response to Reviewer dbgX (4/4)**
>
> > The paper lacks an ethics statement given the use of data involving human subjects, and the nature of the tasks (e.g., age prediction). More details on the used datasets should be added. I did not flag the paper for ethics review yet because I assume that the original datasets included such a statement.
>
> Thank you for pointing this out. **We have added the ethics statements for each dataset in Appendix B.** All of the dataset used in this paper are public datasets. Below we list the ethics statements for each dataset:
>
> - TUAB: This dataset is collected from archival records at Temple University Hospital (TUH). All work was performed in accordance with the Declaration of Helsinki and with the full approval of the Temple University IRB. All personnel in contact with privileged patient information were fully trained on patient privacy and were certified by the Temple IRB. [10]
>
> - MPIIFaceGaze: We have asked the authors of [11] and they confirmed that they have obtained informed consent from all of the participants.
>
> - AgeDB / IMDB / SkyFinder: Those datasets are collected without including any interaction or intervention with human subjects and do not contain any private information [12, 13, 14].
>
> [10] Obeid et al. The Temple University Hospital EEG Data Corpus. Front Neurosci. 2016.
>
> [11] Zhang et al. It’s Written All Over Your Face: Full-Face Appearance-Based Gaze Estimation. CVPR 2017 workshop.
>
> [12] Moschoglou et al. AgeDB: the first manually collected, in-the-wild age database. CVPR 2017 workshop.
>
> [13] Rothe et al. DEX: Deep EXpectation of apparent age from a single image. ICCV 2015 workshop.
>
> [14] Mihail et al. Sky segmentation in the wild: An empirical study. WACV 2016.
>
> > Tables 1-4 (and also others) are lacking standard deviations / confidence intervals. For readers less familar with the datasets, it is hard to judge whether the performance improvement is signficant in all cases. If the authors want to save compute, at least running the best results for 3-5 seeds would be great.
>
> Thanks for your suggestion. We ran the best results on each dataset with 5 different random seeds and **have added them in Appendix E.4**. The table below shows the average prediction error and standard deviations. Our reported numbers in the paper are aligned with the results below.
>
> | AgeDB - SupCR(L1) | TUAB - SupCR(DLDL-v2) | MPIIFaceGaze - SupCR(OR) | SkyFinder - SupCR(DLDL-v2) |
> |:-------------------:|:-----------------------:|:--------------------------:|:----------------------------:|
> |    6.19 ± 0.08    |      7.00 ± 0.10      |       5.24  ± 0.13       |         2.87 ± 0.04        |
>
> > Details on the grid search are missing (search range, number of samples per hyperparameter, ...)
>
> For the encoder training of our method and regression learning baselines, we picked the best learning rates and weight decays for each dataset by grid search, with a grid of learning rates from {0.01, 0.05, 0.1, 0.2, 0.5, 1.0} and weight decays from {1e-6, 1e-5, 1e-4, 1e-3}. For the predictor training of our method, we add no weight decay to the above choices of weight decays and keep learning rate choices unchanged when doing grid search. **We have added these details in Appendix D**.
>
> > Are all datasets taken from an established benchmark, or is the particular experimental protocol used in the paper a novel contribution? What was the rationale for using these specific prediction tasks for evaluation?
>
> All datasets are taken from established benchmarks and their original papers are cited after the name of each dataset in Section 4 “Datasets”. To evaluate the proposed method, we tried to find several typical regression tasks with publicly available datasets from different fields, input modalities and target dimensionalities. Therefore, the selected datasets span multiple fields (computer vision, healthcare and human-computer interaction), contain different input modalities (images and physiological signals), and include both one-dimensional targets (e.g., age prediction) and multi-dimensional targets (e.g.,  gaze estimation).
>
> > How did you find the optimal temperature parameter?
>
> We searched the temperature parameter from {0.1, 0.2, 0.5, 1.0, 2.0, 5.0} and picked the best, which is 2.0. **We have added these details in Appendix D.**
>
> > It would be good to clarify whether the authors will full open source the code for the experimental setup/evaluation.
>
> Yes, the code will be fully open-sourced upon publication.
>
> We tried our best to address all of the issues raised by the reviewer. We hope that the response satisfies the reviewer, and that the reviewer will raise the score.

---

> ### Author Response · Authors · 2022-11-16
> **Response to Reviewer dbgX (3/4)**
>
> > The positioning in the literature is not accurate, and the paper title is too broad. The authors already cite two papers (and I propose three additional ones below) that perform contrastive learning on continuous labels for the purpose of regresson tasks.
>
> > The idea of "supervised contrastive regression" and its effectiveness has been demonstrated before and is not novel (which is not a weakness in itself - just the presentation thereof is currently problematic). The related work and intro should be adapted/expanded in this regard ... ...
>
> > In the light of this and the other works cited under Related Work already, the paper title is too broad. I find it necessary to specialize the title into something like "Supervised contrastive regression by ranking negative samples" or similar (i.e., name the particular contribution in the title). It should be made clear that the technical contribution of the paper lies in the specific choice of loss function rather than "Supervised Contrastive Regression" in general. In the intro, paragraphs 3 and 4 should also be updated accordingly, which currently incorrectly claim that this paper introduces the notion of contrastive learning with continuous labels.
>
> Thanks for pointing out the missing citations. **We have added them and discussed them in the introduction and related work section**. **We are happy to and have already changed the title** to the more descriptive title: “Supervised contrastive regression with sample ranking”.
>
> We would like to emphasize however that Dufumier et al. [5, 6] and Schneider et al. [7] do not perform “contrastive regression”. Specifically, Dufumier et al. [5, 6] deal with binary *classification* tasks (i.e., Alzheimer's Disease detection, bipolar disorder detection, etc) using a contrastive loss re-weighted by meta-data (e.g., age), and Schneider et al. [7] aims at learning low-dimensional, interpretable *embeddings* to encode behavioral and neural data using a generalized contrastive loss which samples positives and negatives according to behavior or time labels. The continuous meta-data in [7, 8] and the continuous labels in [9] are not the targets in their tasks. None of these papers is doing *regression* tasks.
>
> For the papers that are already cited in the related work, Yu et al. [8] does action quality assessment by regressing the relative scores between two videos, which is not doing *contrastive* learning and thus it is not about “contrastive regression”. Wang et al. [9] does domain adaptation for gaze estimation by adding a contrastive loss term, but the proposed method does not benefit *regression* learning on the source dataset, it only benefits the domain adaptation task.
>
> **Anyway, we are happy to add the modifier to limit the scope of the title; we just wanted to clarify that our intention was never to incorrectly broaden the scope.**
>
> [5] Dufumier et al. Contrastive Learning with Continuous Proxy Meta-data for 3D MRI Classification. MICCAI 2021.
>
> [6] Dufumier et al. Conditional Alignment and Uniformity for Contrastive Learning with Continuous Proxy Labels. NeurIPS 2021 Workshop.
>
> [7] Schneider et al. Learnable latent embeddings for joint behavioral and neural analysis. Arxiv 2022.
>
> [8] Yu et al. Group-aware contrastive regression for action quality assessment. ICCV 2021.
>
> [9] Wang et al 2022. Contrastive Regression for Domain Adaptation on Gaze Estimation. CVPR 2022.

---

> > ### Comment · Reviewer_dbgX · 2022-12-05
> > **Re 3/4**
> >
> > **Thanks a lot for the text updates and updating the paper title to "Supervised contrastive regression with sample ranking".**
> >
> > I wanted to further discuss the prior work section and positioning in the literature. I think it improved, but is still not fully accurate w.r.t. prior work:
> >
> > > We would like to emphasize however that Dufumier et al. [5, 6] and Schneider et al. [7] do not perform “contrastive regression”. Specifically, Dufumier et al. [5, 6] deal with binary classification tasks (i.e., Alzheimer's Disease detection, bipolar disorder detection, etc) using a contrastive loss re-weighted by meta-data (e.g., age), and Schneider et al. [7] aims at learning low-dimensional, interpretable embeddings to encode behavioral and neural data using a generalized contrastive loss which samples positives and negatives according to behavior or time labels. The continuous meta-data in [7, 8] and the continuous labels in [9] are not the targets in their tasks. None of these papers is doing regression tasks.
> >
> > **Regarding [9]**: From Eq. (7) it seems that the labels are used for regularizing the model, which would be roughly what you write in Table 8 as "Regulariziation". Do you agree?
> >
> > **Regarding [7]**: I agree for [5,6], but please note that [7] has explicit results on using contrastive learning for regression tasks in almost every main figure (there might be more in the supplement), e.g.:
> >
> > 1. A few examples I found are Figure 2d (predicted location of the animal, which is a regression task, there is one self-supervised (CEBRA-time) and one supervised variant (CEBRA-behavior) of the algorithm); the lineplot also very clearly shows this.
> > 2. Same in Figure 3(i): "Decoded trajectory of hand position using CEBRA-Behavior trained on active trial with x,y position of hand. ", this is again a regression task.
> > 3. And again similar in Figure 5 for predicting the frames (although here I acknowledge that since the task is to predict a "frame", this could fall under your argument that the task is cast into a classification problem)
> >
> > In at least example 1 and 2, I disagree with the assessment "The continuous meta-data in [7, 8] and the continuous labels in [9] are not the targets in their tasks. None of these papers is doing regression tasks.": In (1), the position information is used for training the model, and is the decoding target. In (2), the x/y location is used for training, and again the decoding target.
> >
> > The setup is hence equivalent to the one proposed in this paper: Train the embedding with a supervised contrastive learning objective suitable for continuous data; then train a readout algorithm on top of this embedding. The ablation in Table 8 is good in this regard, but essentially finds that this scheme (SSL first, then readout on top) is better than joint training.
> >
> > So overall, there is a strong conceptual similarity (SSL with supervised contrastive learning first, then regression model as readout), but the tasks for evaluating are different, and your paper has more insights when it comes to comparing to other regression techniques (than only kNN/naive Bayes in [7]).
> >
> > ---
> >
> > Please let me know your thoughts on this.
> >
> > **I think this does not take away any novelty from your paper given the added clarifications and adaptation of the title, but the related work section should be slightly updated in that regard.**

---

> > > ### Author Response · Authors · 2022-12-06
> > > **Follow-up response to 3/4**
> > >
> > > Thank you for your timely response and further comments.
> > >
> > > **Regarding [9]**: We are very sorry that we mislabeled the citations after the words "The continuous meta-data in" and "the continuous labels in". The original sentence we intended to write is:
> > > > The continuous meta-data in [5, 6] and the continuous labels in [7] are not the targets in their tasks.
> > >
> > > (we will address your concerns about [7] in the next point). This paragraph is not meant to discuss [9] but for [5, 6, 7]. The discussions for [9] are (a detailed version can be found in our revised related work below in the next point):
> > > > ....Wang et al. [9] does domain adaptation for gaze estimation by adding a contrastive loss term, but the proposed method does not benefit regression learning on the source dataset, it only benefits the domain adaptation task.
> > >
> > > **Regarding [7]**: We thank the reviewer for your further pointing out and agree that [7] has used contrastive learning for regression tasks. We will **update the related work in the revised version as follows** (and also update the introduction accordingly):
> > > > *A couple of recent papers have used the concept of ``contrastive'' or contrastive learning in the context of continuous labels. In particular, [8] learns a model for action quality assessment by regressing the relative scores between two input videos. This work differs from ours in that it does not use the standard contrastive learning framework. [9] proposes to improve domain adaptation for gaze estimation by adding a contrastive loss term to the L1 loss. They show that their approach is beneficial for adapting a gaze estimation model from one dataset (i.e., one domain) to another, but the approach produces no benefits and even reduces performance for the source dataset. In contrast, our approach improves performance for the source dataset as opposed to being specific to domain adaptation. Besides, [5, 6] use a contrastive loss re-weighted by continuous meta-data but the meta-data is not the target and they are working on classification problems. [7] learns low-dimensional and interpretable embeddings to encode behavioral and neural data using a generalized contrastive loss which samples positives and negatives according to continuous behavior or time labels, and shows that the learned embeddings can be used to decode continuous or discrete targets. Our work differs from it in two aspects: 1) the loss function design is different, where the loss in [7] is a direct adaptation from the contrastive loss while our loss incorporates the intrinsic property of regression problems (i.e., the ordered relationships between samples) into the loss design; 2) we focus on more general regression tasks and comprehensively benchmark with previous regression learning techniques across a wide range of tasks and domains, whereas [7] only focuses on behavioral and neural analysis applications.*
> > >
> > > Since currently the paper pdf cannot be uploaded anymore, we plan to **incorporate the updates into the paper upon publication**. Please feel free to let us know if you have any comments.
> > >
> > > We thank the reviewer again for your time and effort in engaging with us and for your helpful feedback. We are more than willing to provide more clarifications.

---

> > > ### Author Response · Authors · 2022-12-07
> > > **Looking forward to your feedback**
> > >
> > > Dear Reviewer dbgX,
> > >
> > > Thank you for your valuable feedback. We have provided the response and revised the related work section according to your suggestions. Since the discussion period is ending in less than a week, if we have successfully addressed all your concerns, then could we respectfully request the reviewer to further update the reviews and raise the score?
> > >
> > > Thanks,
> > >
> > > Authors

---

> ### Author Response · Authors · 2022-11-16
> **Response to Reviewer dbgX (2/4)**
>
> > Since all results are for visual data and the method uses augmentations for training the embedding, I am missing a comparison to state of the art representation learning methods like DINO (Caron et al., 2021) or also older methods such as SimCLR (Chen et al., 2020). Depending on the results, these do not necessarily need to go into the main paper, but it would be good to confirm that the boost in performance can actually be attributed to the supervised contrastive learning scheme, rather than the augmentations. The comparison could be executed within the proposed evaluation protocol (train the model on the data using the respective augmentations, then evaluate - this has been partially done in Table 7 for one of the datasets) and beyond it using the "extra" ImageNet-pretraining data (use the pre-trained checkpoint, evaluate the feature space). I am happy to further discuss the specifics of this experiment before the authors execute it to ensure that the protocol makes sense.
>
> Thanks for the detailed suggestions. Here we show the results of the comparison to representation learning methods, where prediction error (i.e., MAE for AgeDB, TUAB and SkyFinder and angular error for MPIIFaceGaze) is used as the metric, and we use ResNet-50 as the backbone for image datasets (i.e., AgeDB, MPIIFaceGaze, SkyFinder) to compare with ImageNet pre-trained ResNet-50 encoder and use L1 loss as the regression loss. The SimCLR and DINO ResNet-50 ImageNet-pretrained encoders are 800-epoch checkpoints and their ImageNet linear evaluation accuracies are 69.1% and 75.3% respectively. For DINO-related methods, we report both linear evaluation (left) and kNN evaluation results (right, we choose k=20) (separated by “/”). Due to the time limit, here we only finished the experiments of (1)(2) as you suggested and we will update the full results (including (3)) in the paper later.
>
> The results show that SupCR **outperforms all these baselines on all datasets**. Considering that we apply the same augmentations to all methods (including our method, SSL methods, regression-learning baselines), these results further confirm that the performance gain should be **attributed to our proposed supervised contrastive regression scheme** rather than the augmentations.
>
> |                                          |     AgeDB    |      TUAB     |  MPIIFaceGaze |  SkyFinder  |
> |------------------------------------------|:------------:|:-------------:|:-------------:|:-----------:|
> | SimCLR (pre-trained on ImageNet)           |     15.32    |      N/A      |     16.11     |     5.03    |
> | DINO (pre-trained on ImageNet)             | 9.14 / 11.82 |      N/A      | 12.44 / 12.84 | 4.62 / 3.64 |
> | SimCLR (pre-trained on regression dataset) |     9.38     |     11.01     |      9.13     |     4.55    |
> | DINO (pre-trained on regression dataset)   | 9.91 / 12.03 | 11.62 / 12.32 | 11.90 / 12.50 | 5.00 / 4.20 |
> | Supervised L1 loss                       |     6.49     |      7.96     |      5.74     |     2.88    |
> | SupCR(L1)                                |     **6.10**     |      **6.97**     |      **5.16**     |     **2.78**    |
>
> **We have added these experiments in Appendix E.3.**

---

> > ### Comment · Reviewer_dbgX · 2022-11-27
> > **Re: Response 2/4 (additional baselines)**
> >
> > Thanks for adding these results. One remaining limitation here is that the ResNet50 DINO variant is not that strong; have you tried using the the Vision Transformer models? Your result suggests that you can use the pre-trained variant rather than finetuning additionally on your dataset, so this should be quick to evaluate for all available DINO models, including the larger ones.

---

> > > ### Author Response · Authors · 2022-11-28
> > > **Follow-up response (additional baselines)**
> > >
> > > Thank you for your feedback. As you suggested, we further evaluate ViT-B/8 (the best and largest backbone in the [DINO official repo](https://github.com/facebookresearch/dino), ImageNet linear evaluation accuracy: 80.1\%) DINO pre-trained encoder on all visual regression datasets. The results are shown in the table below.
> > >
> > > |                                          |    Params | AgeDB    |  MPIIFaceGaze |  SkyFinder  |
> > > |------------------------------|:------------:|:------------:|:-------------:|:-----------:|
> > > | ResNet-50 DINO (pre-trained on ImageNet)             | 23M| 9.14 / 11.82 | 12.44 / 12.84 | 4.62 / 3.64 |
> > > | ViT-B/8 DINO (pre-trained on ImageNet)             | 85M | 8.39 / 11.43 | 11.22 / 12.29 | 4.30 / 3.48 |
> > > | ResNet-50 SupCR(L1)                                |     23M | **6.10**     |     **5.16**     |     **2.78**    |
> > >
> > > We can see that using a stronger pre-trained backbone for DINO can deliver better performance but is still far inferior to our method. We want to emphasize that our SupCR is tailored for *regression* tasks by capturing the intrinsic ordered relationships between samples, while current state-of-the-art representation learning methods (e.g., DINO, SimCLR) fail to do so. This is the reason why SupCR outperforms these methods by a large margin even if they are using a stronger architecture.
> > >
> > > We believe the above results further highlight the novelty and significance of our method. We thank the reviewer for your time and effort in reviewing our work and we are also glad to see that you are satisfied with Response 1/4 and hope that you can kindly consider raising the score if our additional results and the following responses ([2](https://openreview.net/forum?id=_QZlje4dZPu&noteId=gIk7t-ZrCG),[3](https://openreview.net/forum?id=_QZlje4dZPu&noteId=MegNyb1HHa),[4](https://openreview.net/forum?id=_QZlje4dZPu&noteId=ZmAPyjdzA-)) address your concerns.

---

> > > > ### Comment · Reviewer_dbgX · 2022-12-05
> > > > **Re: Follow-up response (additional baselines)**
> > > >
> > > > Great, thanks for adding these.

---

> ### Author Response · Authors · 2022-11-16
> **Response to Reviewer dbgX (1/4)**
>
> Dear Reviewer dbgX,
>
> Thanks for your swift response, detailed comments, and insightful feedback. Below, we provide additional results and clarifications to address your concerns one by one.
>
> > The models used for the regression tasks are relatively small. Do the results still hold when a more powerful backbone architecture (e.g., a ResNet50, or even more recent architectures) are used? What was the motivation for using a ResNet18?
>
> Thanks for pointing this out. We use a ResNet-18 as the backbone for the three vision datasets (AgeDB, MPIIFaceGaze and SkyFinder) just for saving computation time. Given the limited time and computation resources during the rebuttal period, we computed the first row of Table 1, 3 and 4 with ResNet-50 below (metric: prediction error / coefficient of determination) to show that the results do not change with ResNet-50. We will re-run all experiments with ResNet-50 and add the results to the final paper.
>
> |              |      L1      |   SupCR(L1)  |
> |--------------|:------------:|:------------:|
> | AgeDB        | 6.49 / 0.830 | 6.10 / 0.851 |
> | MPIIFaceGaze | 5.74 / 0.748 | 5.16 / 0.819 |
> | SkyFinder    | 2.88 / 0.863 | 2.78 / 0.877 |
>
> > The evaluation against the baselines is thorough, but I am missing a comparison to conceptually different methods. Even if the proposed method does not outperform them, it is good to provide this as a reference. Can the authors comment on the SOTA models for each of the five benchmarks? E.g. for Table 3, I find a reference result of around 3-5 angular error, while authors report >5. I would be interested in learning about the state of the art of each of the benchmarks (I am not experienced with these datasets, so maybe the tables already contain the applicable SOTA).
>
> Thank you for raising this great point. We **have added SOTA references and included comparison to them in Appendix E.2.** To the best of our knowledge, the SOTA on each dataset are as follows:
>
> - $\texttt{AgeDB}$: Yang et al. [1]. The results reported in their paper is MAE = 7.47, but they use a different optimizer and train the model with fewer epochs. We implemented their method and the result under our evaluation protocol is MAE = 6.58, which is close to the baselines used in our paper. The results of our method is MAE=6.14.
>
> - $\texttt{TUAB}$: Engemann et al. [2]. They use an L1 loss to train the regression model and results reported in their paper are MAE = 7.75 and R^2 = 0.60, but they use a different backbone model from us. The results of our implementation of L1 loss reported in our paper are MAE = 7.96 and R^2 = 0.599, which is close to their results. The results of our method are MAE=6.91 and R^2=0.697.
>
> - $\texttt{MPIIFaceGaze}$:  Abdelrahman et al. [3]. Since we use a subsampled version of MPIIFaceGaze dataset to save computation time and also split the training set and test set by ourselves, the numbers reported in their paper are **not directly comparable** to our results. We implemented their method and the result under our evaluation protocol is angular error = 5.76. They use a classification-based method to do gaze estimation, and the result is close to the classification-based regression baselines (DEX angular error = 5.72 and DLDL-v2 angular error = 5.47) selected in our paper. The results of our method is angular error = 5.13.
>
> - $\texttt{SkyFinder}$: Chu et al. [4]. They use a classification-based method with reported performance RMSE=4.28. We implemented their method under our experimental settings and the results are RMSE=4.05 and MAE=2.97, which is very close to the classification-based regression baseline (DLDL-v2) selected in our paper. The results of our method are RMSE = 3.84 and MAE=2.85.
>
> - $\texttt{IMDB}$: Since we use this dataset just for evaluating several properties of the proposed method, we subsampled this dataset according to the need of each experiment. Therefore, the results on this dataset are just for comparing with the baseline and are not suitable for comparing with SOTA.
>
> Overall, our method **surpasses the state-of-the-art methods on all benchmarks**. We summarize the above comparisons in the following table.
>
> |              |     Metric    |       SOTA       |   Ours  |
> |--------------|:-------------:|:----------------:|:------------:|
> | AgeDB        |      MAE      |     6.58 [1]     |     **6.14**     |
> | TUAB         |   MAE / R^2   | 7.96 / 0.599 [2] | **6.91 / 0.697** |
> | MPIIFaceGaze | Angular Error |     5.76 [3]     |     **5.13**     |
> | SkyFinder    |   MAE / RMSE  |  2.97 / 4.05 [4] |  **2.85 / 3.84** |
>
> [1] Yang et al. Delving into Deep Imbalanced Regression. ICML 2021.
>
> [2] Engemann et al. A reusable benchmark of brain-age prediction from M/EEG resting-state signals. NeuroImage 2022.
>
> [3] Abdelrahman et al. L2CS-Net: Fine-Grained Gaze Estimation in Unconstrained Environments. Arxiv 2022.
>
> [4] Chu et al. Visual weather temperature prediction. WACV 2018.

---

> > ### Comment · Reviewer_dbgX · 2022-11-27
> > **Re: Response 1/4**
> >
> > > Re ResNet50:
> >
> > Thanks for adding these. The results address my concern, the improvement still holds.
> >
> > > Comparisons to SOTA
> >
> > Thanks for adding this, good addition. Great to see you outperform the respective SoTA.

---

> ### Author Response · Authors · 2022-12-05
> **We would like to hear back from Reviewer dbgX**
>
> Dear Reviewer dbgX,
>
> Thank you for your prompt replies and thoughtful discussions with us back and forth, which greatly helped improve the quality of our work. We have provided the results of additional baselines for your latest comments. We believe these new results together with previous results and clarifications can address all your concerns. We are also very happy to discuss and provide further clarifications.
>
> > I made specific suggestions on how to improve these weaknesses above, and am willing to adapt my scores.
>
> As you mentioned in your initial review, considering that the discussion deadline is approaching (we sincerely remind you that the phase of the discussion period ends on Dec. 12, i.e., next Monday), and that we have addressed all your concerns, could you please kindly consider updating the reviews and raising your score in light of our responses?
>
> Thanks,
>
> Authors

---

> > ### Comment · Reviewer_dbgX · 2022-12-09
> > **I will raise my score, thanks for the updates!**
> >
> > Dear authors,
> >
> > thanks again for the good discussion and improvements you added to the paper. I will raise my score, but am still undecided by how much. I would like to take the last days of the discussion period to sync with the other reviewers and the AC if I might have missed something in my assessment, but it will def. move towards acceptance (i.e., >= 6).
> >
> > I already updated the score to 6.

---

> > > ### Author Response · Authors · 2022-12-11
> > > **Thanks for your updates!**
> > >
> > > Thanks for your updates! We are delighted to see that our responses adequately address your concerns. Please feel free to let us know if there are any other clarifications we can offer before the discussion period ends.
> > >
> > > Once again, many thanks for your time and dedication to the review process and helpful discussions with us back and forth, we are extremely grateful.

---

### Official Review · Reviewer_hEGc · 2022-10-26

**Confidence:** 3
**Correctness:** 3
**Technical Novelty And Significance:** 3
**Empirical Novelty And Significance:** 3
**Recommendation:** 6

**Clarity, Quality, Novelty And Reproducibility:**

The clarity needs some improvement, which can be seen in the following Summary Of The Review in detail.
The quality and novelty of the work are high, as it proves the error bound where the embedding guarantees that the order of similarity is consistent with the order of the similarity of labels.



**Details Of Ethics Concerns:**

na.

**Strength And Weaknesses:**

The paper is written clearly and organized well. Contrastive learning for regression is an interesting topic where deep learning does not well addressed fully. Experiments show noticeable improvement over the baseline in terms of accuracy/loss.
Weakness: 1. Please clarify how to determine the anchors in a batch; or the authors just enumerate all samples and assign each sample as an anchor?
2. The proof needs more explanation step by step in the Appendix. The theorem guarantees that the order of similarity is consistent with the order of the similarity of labels. But it is not necessary that the order can preserve the continuity and the specific distance between labels. I am curious if the authors can make some more investigation into this concern.

**Summary Of The Paper:**

This paper attempts to incorporate Contrastive Learning into regression problems. The format of contrastive loss is not changed compared with SupCon, except for the definition of positive/negative pairs within the dataset. Namely, an anchor is first defined as a landmark and
 the sample with the closest label to the anchor's label is then defined as the corresponding positive sample and other remaining samples in a mini-batch are negative samples. optimizing LSupCR will make the feature embedding ordered according to the order in the label space. This definition of positivity/negativity is used to make the feature embedding ordered according to the order in the label space.


**Summary Of The Review:**

Here are some concerns that might need clarification:
1. How do authors choose the anchors in a batch? Is the anchor defined randomly, or every sample has to be enumerated?
2. How to obtain the last inequality from the second last inequality of Eq. (2)? Namely, how to derive Eq (3)?
3. How to obtain Eq. (4)? There are many long equations that lack detailed explanation, which makes it hard to parse, although the proof is placed in the Appendix. Please add the necessary explanation of the derivation step by step.
4. In SupCon and other related works, the feature embeddings are often normalized of length 1. In this case, the maximum distance between two samples is 2, which seems to enforce the delta in Def 1 to be greater than 0.5. This restriction may affect the feasible range of epsilon. I am wondering if the authors thought about this.
5. In the experiments, the performance suggests that SUPCR is more robust to unforeseen data corruption. I am curious if that's because the data augmentation is used as much contrastive learning did? Would the authors want to discuss this?
6. To obtain the smoothness and continuity of Figure 1 c, do the authors have to choose many anchors so that the embeddings can be less discrete like how SupCon shows in Figure b? So how is the training efficiency of the model?

---

> ### Author Response · Authors · 2022-11-16
> **Response to Reviewer hEGc (2/2)**
>
> > In the experiments, the performance suggests that SUPCR is more robust to unforeseen data corruption. I am curious if that's because the data augmentation is used as much contrastive learning did? Would the authors want to discuss this?
>
> As mentioned in the experiment setup, the **same** data augmentations are used for the baselines and our method. In the data corruption experiments, the L1 model is trained with the same data augmentation as SupCR(L1): random crop and resize (with random flip) and color distortions. Besides, We use the corruption generation process in [Hendrycks & Dietterich, 2019](https://github.com/hendrycks/robustness) to generate the data corruptions, including the noise, blur and weather filters (snow, frost, fog). These types of corruptions cannot be simply circumvented by data augmentation since they are not correlated with each other, but require the model’s representation to have inherent smoothness and robustness against the data corruption.
>
> > To obtain the smoothness and continuity of Figure 1 c, do the authors have to choose many anchors so that the embeddings can be less discrete like how SupCon shows in Figure b? So how is the training efficiency of the model?
>
> We would like to clarify that the number of *anchors* is the same for SupCR (our proposed method) and SupCon, which is the number of all samples in the two-view batch, i.e., 2N. However, the number of samples to be treated as *positive* samples for each anchor is different between SupCR and SupCon, where all samples in the batch (i.e., 2N) in SupCR will be treated as the positive while only samples with the same label as the anchor will serve as the positive in SupCon. This makes the time cost for loss computation in SupCR a bit higher than that for SupCon. However, the time cost for loss computation is relatively small in comparison to other time costs during training (e.g.,data augmentation, model forwarding, gradient back-propagation, etc) for both SupCR and SupCon.  So the training efficiency of SupCR stays comparable to that of SupCon.
>
> We compute the average wall clock running time (in seconds) per training epoch on 8 NVIDIA TITAN RTX GPUs for SupCon and SupCR on each of the four datasets, shown in the following table. Results show that the training efficiency of SupCR is comparable to the SupCon.
>
> |              | SupCon | SupCR | ratio |
> |--------------|:--------:|:-------:|:-------:|
> | AgeDB        | 23.1   | 26.2  | 1.13x |
> | TUAB         | 25.3   | 27.3  | 1.08x |
> | MPIIFaceGaze | 69.1   | 75.4  | 1.09x |
> | SkyFinder    | 55.6   | 61.8  | 1.11x |
>
> **We have added the experiments about training efficiency in Appendix E.6.**

---

> ### Author Response · Authors · 2022-11-16
> **Response to Reviewer hEGc (1/2)**
>
> Dear Reviewer hEGc,
>
> Thank you very much for acknowledging the quality and the novelty of our work. In the following, we address your concerns one by one. We hope that these will further clarify our work and lead to a favorable increase of the score.
>
> > (Summary) Namely, an anchor is first defined as a landmark and the sample with the closest label to the anchor's label is then defined as the corresponding positive sample and other remaining samples in a mini-batch are negative samples.
>
> Here, we would like to clarify a potential misunderstanding. In the proposed method, and for each anchor, not only the nearest sample will be treated as positive, but each of the samples in the batch (i.e., the closest, second closest, third closest, etc.) will each be considered as a positive sample; but each of these positive samples has different corresponding negative samples, which are the samples that are farther from the anchor than the positive sample. For example, for an anchor, when the positive sample is the closest sample in the batch to the anchor, the negative samples are the second closest sample to the anchor, the third closest, the fourth closest, etc.
>
> >  Please clarify how to determine the anchors in a batch; or the authors just enumerate all samples and assign each sample as an anchor?
>
> > How do authors choose the anchors in a batch? Is the anchor defined randomly, or every sample has to be enumerated?
>
> Similar to other contrastive learning frameworks, e.g. SimCLR, SupCon, all samples in the batch are enumerated as the anchor. As shown in equation (1), i is the enumerator of the anchor and it enumerates from 1 to 2N.
>
> > The proof needs more explanation step by step in the Appendix.
>
> > There are many long equations that lack detailed explanation, which makes it hard to parse, although the proof is placed in the Appendix. Please add the necessary explanation of the derivation step by step.
>
> We apologize for the potential confusion caused by the limited explanation in the Appendix and thank the reviewer for the suggestion. **We have broken down the long equations and added the corresponding explanation to the proof**.
>
> > The theorem guarantees that the order of similarity is consistent with the order of the similarity of labels. But it is not necessary that the order can preserve the continuity and the specific distance between labels. I am curious if the authors can make some more investigation into this concern.
>
> It is intentional that loss function guarantees the order of feature distances is consistent with the order of label distances and does *not* impose constraints on the specific feature distances. We would like to allow for a general regressor that may be nonlinear, i.e., the relationship between the feature space and the label space need not be linear. Imposing the exact distance between labels on features can lead to suboptimal results in the case of nonlinearity, whereas imposing the same ordering while allowing for differences in the exact distances provides more flexibility.
>
> > How to obtain the last inequality from the second last inequality of Eq. (2)?
>
> The last inequality of Eq.(2) in the original submission (Eq.(4) in the current revised version) comes from Jensen's Inequality. In Eq.(3), the terms after $\frac{1}{n_{i, m}}$ in the $\log$ sum up to 1 since the sum of numerators is equal to the denominator.
>
> > How to derive Eq (3)?
>
> For Eq.(3) in the original submission (Eq. (5) in the current revised version), it is because of the following: For such a set of feature embeddings, $s_{i,j} = s_{i,k}$ when $d_{i,j} = d_{i,k}$, so, in the $\log$, all $s_{i,k}$ in the denominator are equal to the $s_{i, j}$ in the numerator, and thus the denominator is $n_{i,m}$ times of the numerator.
>
> > How to obtain Eq. (4)?
>
> For Eq.(4) in the original submission (Eq. (6) in the current revised version), it is because all $(s_{i,k} − s_{i,j})$ in the numerator $< -\gamma$ and all $(s_{i,k} − s_{i,j})$ in the denominator $= 0$.
>
> > In SupCon and other related works, the feature embeddings are often normalized of length 1. In this case, the maximum distance between two samples is 2, which seems to enforce the delta in Def 1 to be greater than 0.5. This restriction may affect the feasible range of epsilon. I am wondering if the authors thought about this.
>
> Yes, we have considered this. Since SupCon and other related works use *cosine similarity* (the dot product after normalization) as the feature similarity measure, the feature similarity ranges from -1 to 1 and the maximum feature similarity gap is 2. However, we can use feature similarity measures that do not constrain the maximum similarity gap in the loss function. For example, in our implementation, as mentioned in experiment settings, we use *negative L2 norm* as the feature similarity measure, so the feature embeddings are not normalized, the feature similarity ranges from -infinity to 0 and the similarity gap can be arbitrarily large.

---

> ### Author Response · Authors · 2022-11-29
> **We would like to hear back from Reviewer hEGc**
>
> Dear Reviewer hEGc,
>
> Thank you for your time and effort in reviewing our work. We believe we have addressed all your concerns and questions. We are happy to discuss and provide further clarifications. Considering that the discussion deadline is approaching, could you please kindly consider updating the reviews and raising your score in light of our response?
>
> Thanks,
>
> Authors

---

> > ### Author Response · Authors · 2022-12-11
> > **Discussion period ending soon; We would like to hear back from Reviewer hEGc**
> >
> > We would like to thank the reviewer again for your time and effort. We have provided additional experiments, results, and clarifications to respond to your comments. We believe these should have successfully addressed all your concerns.
> >
> > Given there are only ***2 days left*** in the discussion period, we wanted to check if the reviewer had seen our responses and whether there were any more clarifications the reviewer would like. If the concerns of the reviewer are clarified and the reviewer is convinced of the novelty, significance, and completeness of our work, we'd be grateful if the reviewer could update your review and increase your score to reflect that, so we know that our response has been seen.
> >
> > Thanks, Authors

---

### Official Review · Reviewer_ZitN · 2022-11-02

**Confidence:** 3
**Correctness:** 3
**Technical Novelty And Significance:** 3
**Empirical Novelty And Significance:** 3
**Recommendation:** 3

**Clarity, Quality, Novelty And Reproducibility:**

The text is clear, experiment details are lacking, e,g,
- how was the grid search setup for both baselines and current proposed approach?
- Does L1 refer to the same base model but trained solely with L1 loss?
- How does the loss behave through out training? I'd presume mode collapse could easily happen for many cases where the augmentation distorts the input but maintains the same target y.


**Strength And Weaknesses:**

The overall idea somewhat makes sense, but some parts are lacking.
- Most datasets selected are image datasets, though often regression tasks are on time-series/tabular data or other data modalities where creating "augmented" views drastically distorts the input where the label is no longer a valid one. It needs to be clarified how this would work for such datasets, where the relationship between augmenting/transforming the input and its correspondence on the target.
- on the TUAB dataset, what augmentations were used? How is random-crop/random flip related to age? Is the same age used for both samples? In that case, augmented views of the sample will always have label distance 0, even though the augmentation may completely distort the input.
- What happens if the target is cyclical? This hierarchical setup for contrasting iteratively will fail for such cases unless I misunderstand parts of the algorithm.
- What happens to the contrastive term on the furthest sample? Is this dropped? I'm assuming it is.
- Alternate baselines that pre-train or use an SSL loss as a regularizer along with the supervised L1 loss term should be added. For example, why is this method is compared with CORN?

**Summary Of The Paper:**

The authors proposed a supervised contrastive regression loss for regression tasks, different from the supervised contrastive loss used for classification. The loss term is the same as NT-Xent where in their proposed approach, positive samples of an augmented view must fall within a radius or distance in the (continuous) label space, and negatives fall outside of this distance. Subsequently, they iteratively apply this term for the first closest positive sample, second, third and so on until the N-1 farthest sample. They use this to pre-train networks, which are further fine-tuned by freezing the model and training a single linear head, as in self-supervised methods.

**Summary Of The Review:**

Overall, the applicability of this approach on other datasets is not clear and although the results do seem promising it's hard to assess whether the gain is simply coming from pre-training.

---

> ### Author Response · Authors · 2022-11-16
> **Response to Reviewer ZitN (3/3)**
>
> > How does the loss behave throughout training? I'd presume mode collapse could easily happen for many cases where the augmentation distorts the input but maintains the same target y.
>
> The loss gradually goes down as training progresses and converges at the end. As discussed in detail above, the augmentations do not distort the input or change the label. Therefore, the training does not suffer from the mode collapse issue. This can also be verified by the linear probing performance, since a collapsed model cannot achieve good performance under a linear probing protocol.
>
> > Overall, the applicability of this approach on other datasets is not clear and although the results do seem promising it's hard to assess whether the gain is simply coming from pre-training.
>
> As discussed in detail in our answer to the first question, our method uses the standard data augmentations used by traditional contrastive learning (e.g., SimCLR) for each dataset and modality. We hope that our answer therein has addressed the concern raised by the reviewer regarding the applicability of our method to other datasets and modalities.
>
> In our paper, we have also shown the applicability of our method to five real-world datasets that span computer vision, human-computer interaction, and healthcare and include images and time series modalities. Therefore, we believe our method is universal and applies to other datasets.
>
> Furthermore,  we have performed the experiments suggested by the reviewer to show that our gains are not from pre-training.  As the table results and discussions above show, the performance gain of our method is from the proposed SupCR loss rather than the pre-training scheme.
>
> We appreciate it if the reviewer can let us know whether our response addresses his/her concerns, and consider improving the score.

---

> ### Author Response · Authors · 2022-11-16
> **Response to Reviewer ZitN (2/3)**
>
> > What happens to the contrastive term on the furthest sample? Is this dropped? I'm assuming it is.
>
> No. For an anchor $i$, let $p$ and $q$ denote the two views of the furthest sample. Then, the last two contrastive terms of the loss will be $-\log \frac{\exp (\operatorname{sim}(\boldsymbol{v}_i, \boldsymbol{v}_p) / \tau)}{\exp (\operatorname{sim}(\boldsymbol{v}_i, \boldsymbol{v}_p) / \tau) + \exp (\operatorname{sim}(\boldsymbol{v}_i, \boldsymbol{v}_q) / \tau)}$ and $-\log \frac{\exp (\operatorname{sim}(\boldsymbol{v}_i, \boldsymbol{v}_q) / \tau)}{\exp (\operatorname{sim}(\boldsymbol{v}_i, \boldsymbol{v}_p) / \tau) + \exp (\operatorname{sim}(\boldsymbol{v}_i, \boldsymbol{v}_q) / \tau)}$. They will make the features of two augmented views of the furthest sample have the same distances to the anchor.
>
> > Alternate baselines that pre-train or use an SSL loss as a regularizer along with the supervised L1 loss term should be added. For example, why is this method is compared with CORN?
>
> We have **already included** the comparison to baselines that use SimCLR and SupCon loss to pre-train the encoder for the AgeDB dataset in Table 7 of the original submission.  The results in that table show that our SupCR loss outperforms those two by a large margin. This is because the SimCLR loss does not utilize label information and SupCon loss fails to leverage the ordered relationship among labels.
>
> Here, following your suggestions, we further run experiments on all datasets using SimCLR loss to pre-train or regularize. We also include experiments that directly evaluate the regression performance on the encoder pre-trained on ImageNet using the SimCLR loss, as suggested by reviewer dbgX. Results are shown in the table below, where the prediction error (i.e., MAE for AgeDB, TUAB and SkyFinder, and angular error for MPIIFaceGaze) is used as the metric, and we use ResNet-50 as the backbone for image datasets (i.e., AgeDB, MPIIFaceGaze, SkyFinder) to compare with ImageNet pre-trained ResNet-50 encoder and use L1 loss as the regression loss.
>
> The results show that SupCR **outperforms all of these baselines on all datasets**, and pre-training using SimCLR or using SimCLR as a regularizer performs worse than the supervised L1 baseline. This verifies that the **performance gain** of our method **stems from our proposed SupCR loss** rather than the pre-training scheme.
>
> |                                            |   AgeDB  |   TUAB   | MPIIFaceGaze | SkyFinder |
> |--------------------------------------------|:--------:|:--------:|:------------:|:---------:|
> | SimCLR (pre-trained on ImageNet)             |   15.32  |    N/A   |     16.11    |    5.03   |
> | SimCLR (pre-trained on regression dataset)   |   9.38   |   11.01  |     9.13     |    4.55   |
> | SimCLR (using as a regularizer with L1 loss) |   6.53   |   8.02   |     5.85     |    2.93   |
> | Supervised L1 loss                         |   6.49   |   7.96   |     5.74     |    2.88   |
> | SupCR(L1)                                 | **6.10** | **6.97** |   **5.16**   |  **2.78** |
>
> **We have added these experiments in Appendix E.3.**
>
> As for the question of why our method is compared with CORN, this is because we would like to show that SupCR can consistently improve previous regression learning methods, i.e., for any method, adding the SupCR loss to learn the representation is beneficial, as shown in Table 1, Table 2, Table 3 and Table 4 in the submission. Therefore, several typical regression learning methods, including CORN, were used as baselines.
>
> > How was the grid search setup for both baselines and current proposed approach?
>
> For the encoder training of our method and regression learning baselines, we picked the best learning rates and weight decays for each dataset by grid search, with a grid of learning rates from {0.01, 0.05, 0.1, 0.2, 0.5, 1.0} and weight decays from {1e-6, 1e-5, 1e-4, 1e-3}. For the predictor training of our method, we add no weight decay to the above choices of weight decays and keep learning rate choices unchanged when doing grid search. **We have added these details in Appendix D.**
>
> > Does L1 refer to the same base model but trained solely with L1 loss?
>
> Yes. L1 means training the entire model (encoder + predictor) end-to-end with L1 loss. SupCR(L1) means first training the encoder with SupCR loss and then freezing it and training the predictor on top of it with L1 loss. The model architecture is the same. We have **updated the text in the first paragraph of Section 4.1 accordingly** to clarify this.

---

> ### Author Response · Authors · 2022-11-16
> **Response to Reviewer ZitN (1/3)**
>
> Dear Reviewer ZitN,
>
> Thanks for your constructive comments and insightful feedback. They are very helpful for further improving the quality of our paper. However, we believe that there are several **important misunderstandings** which we would like to clarify and address here one by one.
>
> > Most datasets selected are image datasets, though often regression tasks are on time-series/tabular data or other data modalities where creating "augmented" views drastically distorts the input where the label is no longer a valid one. It needs to be clarified how this would work for such datasets, where the relationship between augmenting/transforming the input and its correspondence on the target.
>
> First, we would like to clarify that our approach is **orthogonal** to augmentations, and works with the standard augmentations used in contrastive learning for the particular dataset and modality. We have run experiments with both images and time series datasets. In both cases, we used standard augmentations proposed in the literature for each modality. Specifically, as is common in the literature on contrastive learning, for image datasets we used random crop and resize (with random horizontal flip), color distortions [1,2], and for time series datasets, we used random crop [3,4,5].  Also, please note that in our experiments, the same augmentations are used for our approach and the baseline methods.
>
> Second, please note that augmentations in contrastive learning should **never** “drastically distort the input where the label is no longer a valid one”. This is a **general requirement** for augmentations in contrastive learning (otherwise contrastive learning would not work with that modality). For example, when applying contrastive learning to gaze direction detection, one cannot use random horizontal flip since it changes the direction of the gaze. This is true for SimCLR and traditional contrastive learning. Similarly, for EEG signals, one cannot resize the signal or use random flips, even if one is using traditional contrastive algorithms, e.g., SimCLR. Since our approach does not change the proper augmentations for contrastive learning as applied to each modality and dataset, this issue of distortion does not occur with our approach or the baselines.
>
>  In Section 4 “Experiment Setting” in the original submission, we reported the augmentations we use for each dataset:
>
> - AgeDB / IMDB / SkyFinder (image): random crop and resize (with random horizontal flip), color distortions;
>
> - MPIIFaceGaze (image): random crop and resize (without random horizontal flip), color distortions;
>
> - TUAB (EEG signals, time series): random crop.
>
> We **added visualizations of augmentations for each dataset in Appendix C** to make it clearer in the current revised version.
>
> [1] Chen et al. A Simple Framework for Contrastive Learning of Visual Representations. ICML 2020.
>
> [2] He et al. Momentum Contrast for Unsupervised Visual Representation Learning. CVPR 2020.
>
> [3] Franceschi et al. Unsupervised Scalable Representation Learning for Multivariate Time Series. NeurIPS 2019.
>
> [4] Tonekaboni et al. Unsupervised Representation Learning for Time Series with Temporal Neighborhood Coding. ICLR 2021.
>
> [5] Yue et al. TS2Vec: Towards Universal Representation of Time Series. AAAI 2022.
>
> > On the TUAB dataset, what augmentations were used? How is random-crop/random flip related to age? Is the same age used for both samples? In that case, augmented views of the sample will always have label distance 0, even though the augmentation may completely distort the input.
>
> As mentioned above, random crop is used for TUAB (please note that the input is not resized after cropping). We do not use random flip for TUAB dataset. Yes, the same age is used for both samples. Random crop does not distort the input or change the label (i.e., age) because cropped views of signals belong to the same instance. Random crop has been commonly used as augmentation in time-series contrastive learning literatures [3,4,5].
>
> > What happens if the target is cyclical? This hierarchical setup for contrasting iteratively will fail for such cases unless I misunderstand parts of the algorithm.
>
> Our method is applicable to the case where the target is cyclical by designing a cyclical label distance function $d(\cdot,\cdot)$. For example, considering the task where the target is the angle degree $\in [0, 360)$, a cyclical label distance function can be expressed as $d(y_1, y_2) = \min(|y_1-y_2|, 360-|y_1-y_2|)$. In this case, for the sample with label 0, the sample with label 1 and the sample with 359 will have the same distance, which is 1, to it.

---

> ### Author Response · Authors · 2022-11-29
> **We would like to hear back from Reviewer ZitN**
>
> Dear Reviewer ZitN,
>
> We would like to thank the reviewer for the great questions. We believe our additional experiments, results, and clarifications (Response [1](https://openreview.net/forum?id=_QZlje4dZPu&noteId=wtel4VB60C), [2](https://openreview.net/forum?id=_QZlje4dZPu&noteId=cguatXFeGB7), [3](https://openreview.net/forum?id=_QZlje4dZPu&noteId=_CGZ9-h5aO)) have successfully addressed your misunderstandings and concerns.
>
> If the concerns of the reviewer are clarified and the reviewer is convinced of the novelty, significance, and completeness of our work, then can we respectfully request the reviewer to increase his/her score from 3?
>
> Thanks,
>
> Authors

---

> > ### Author Response · Authors · 2022-12-11
> > **Discussion period ending soon; We would like to hear back from Reviewer ZitN**
> >
> > We would like to thank the reviewer again for your time and effort. We have provided additional experiments, results, and clarifications to respond to your comments. We believe these should have successfully addressed all your misunderstandings and concerns.
> >
> > Given there are ***only 2 days left*** in the discussion period, we wanted to check if the reviewer had seen our responses and whether there were any more clarifications the reviewer would like. If the concerns and misunderstandings of the reviewer are clarified and the reviewer is convinced of the novelty, significance, and completeness of our work, we'd be grateful if the reviewer could update your review and increase your score to reflect that, so we know that our response has been seen.
> >
> > Thanks, Authors

---

### Author Response · Authors · 2022-11-19
**General Response**

Dear Reviewers,

Thank you for your valuable time and effort. We are glad to see that you found:
- the paper is clearly-written and well-organized (ZitN, hEGc, dbgX, 7jug);
- the proposed method is novel (hEGc, dbgX) and simple to implement (dbgX);
- the experiments are comprehensive (dbgX, 7jug) and the improvements are noticeable (hEGc).

We have provided responses to each of the concerns you raised and have revised the paper (changes are highlighted in red) according to the suggestions. We hope our responses adequately address your questions and we welcome any further comments and discussions. We would really appreciate it if you could consider raising your ratings after reading our responses.

Thanks,

Authors

---

### Author Response · Authors · 2022-11-24
**Looking forward to your feedback**

Dear Reviewers,

Thank you for your valuable comments and suggestions, which greatly helped improve our paper.

We have provided detailed responses and revised the paper accordingly, but have not received any responses yet.

Please do not hesitate to let us know if there are any additional clarifications or experiments we can offer.

Best,

Authors

---

### Author Response · Authors · 2022-12-08
**General Response to All Reviewers and AC**

Dear Reviewers and AC,

Thank you for your constructive comments and insightful questions, which are very helpful in improving our paper. To address your concerns, we have answered all of the questions and made the following changes to the paper:

- Incorporated the writing and reference suggestions in **Sections 1, 2, 4** (Reviewer [ZitN](https://openreview.net/forum?id=_QZlje4dZPu&noteId=cguatXFeGB7), [dbgX](https://openreview.net/forum?id=_QZlje4dZPu&noteId=MegNyb1HHa));

- Elaborated the explanations for the proof in **Appendix A** (Reviewer [hEGc](https://openreview.net/forum?id=_QZlje4dZPu&noteId=xK460qTyFr-));

- Added ethics statements in **Appendix B** (Reviewer [dbgX](https://openreview.net/forum?id=_QZlje4dZPu&noteId=ZmAPyjdzA-));

- Added data augmentation examples for each dataset in **Appendix C** (Reviewer [ZitN](https://openreview.net/forum?id=_QZlje4dZPu&noteId=wtel4VB60C));

- Added more experimental details in **Appendix D** (Reviewer [ZitN](https://openreview.net/forum?id=_QZlje4dZPu&noteId=cguatXFeGB7), [dbgX](https://openreview.net/forum?id=_QZlje4dZPu&noteId=ZmAPyjdzA-), [7jug](https://openreview.net/forum?id=_QZlje4dZPu&noteId=p2S2qXWDln9));

- Added experiments with larger backbone models on each vision dataset in **Appendix E.1** (Reviewer [dbgX](https://openreview.net/forum?id=_QZlje4dZPu&noteId=rKXYInfZtQ));

- Added comparisons to existing SOTA methods on each dataset in **Appendix E.2** (Reviewer [dbgX](https://openreview.net/forum?id=_QZlje4dZPu&noteId=rKXYInfZtQ));

- Added comparisons to SOTA self-supervised learning methods in **Appendix E.3** (Reviewer [ZitN](https://openreview.net/forum?id=_QZlje4dZPu&noteId=cguatXFeGB7), [dbgX](https://openreview.net/forum?id=_QZlje4dZPu&noteId=gIk7t-ZrCG));

- Added standard deviations of results in **Appendix E.4** (Reviewer [dbgX](https://openreview.net/forum?id=_QZlje4dZPu&noteId=ZmAPyjdzA-));

- Added ablation on numbers of positives of the loss function in **Appendix E.5** (Reviewer [7jug](https://openreview.net/forum?id=_QZlje4dZPu&noteId=p2S2qXWDln9));

- Added results about training efficiency in **Appendix E.6** (Reviewer [hEGc](https://openreview.net/forum?id=_QZlje4dZPu&noteId=z1OGcrRXTDF)).

We sincerely remind you that the discussion period is **only 5 days left** and we have heard back from ***only one reviewer*** till now. We thank all reviewers again for your time and feedback. We hope our responses have adequately addressed your questions. We are also very happy to discuss and provide further clarifications.

If we have addressed your concerns, could you please kindly consider updating the reviews and raising your scores in light of our responses and efforts?

Thanks,  Authors

---

### Decision · Program_Chairs · 2023-01-20

**Decision:**

Reject

**Justification For Why Not Higher Score:**

Limited  technical novelty

**Justification For Why Not Lower Score:**

N/A

**Metareview: Summary, Strengths And Weaknesses:**

This work presents a contrastive loss for regression tasks. The format of contrastive is based on prior work for contrastive learning for classification, but the definition of positive/negative pairs are adapted to the regression task. The paper presents a theoretical justification for their loss function showing that the resulting feature embedding ordering will be consistent with the ordering of the regression target.
Experiments show improvements of their method both in terms of accuracy and robustness to image perturbations on a range of different evaluation tasks.

Strength:
+ regression task has received limited attention for representation learning, this work will be a nice addition, despite its simplicity
+ demonstrated effectiveness for regression task on data for which reliable augmentation methods exist
+ the method is simple, easy to apply, and is supported by theoretical justification
+ the experimental results seem convincing, especially with the additional ablation studies suggested by the reviewers

Weakness:
- technical novelty is limited
- rely on data augmentation --- not applicable for domains with no appropriate data augmentation methods


**Summary Of Ac-Reviewer Meeting:**

Discussion via email asynchronously in lieu of a virtual meeting, to confirm the strengths and weaknesses of the work.